# Miro proteins coordinate microtubule- and actin-dependent mitochondrial transport and distribution

Guillermo López-Doménech [iD], Christian Covill-Cooke, Davor Ivankovic, Els F Halff, David F Sheehan, Rosalind Norkett, Nicol Birsa & Josef T Kittler* [iD]

## Abstract

In the current model of mitochondrial trafficking, Miro1 and Miro2 Rho-GTPases regulate mitochondrial transport along microtubules by linking mitochondria to kinesin and dynein motors. By generating Miro1/2 double-knockout mouse embryos and single- and double-knockout embryonic fibroblasts, we demonstrate the essential and non-redundant roles of Miro proteins for embryonic development and subcellular mitochondrial distribution. Unexpectedly, the TRAK1 and TRAK2 motor protein adaptors can still localise to the outer mitochondrial membrane to drive anterograde mitochondrial motility in Miro1/2 double-knockout cells. In contrast, we show that TRAK2-mediated retrograde mitochondrial transport is Miro1-dependent. Interestingly, we find that Miro is critical for recruiting and stabilising the mitochondrial myosin Myo19 on the mitochondria for coupling mitochondria to the actin cytoskeleton. Moreover, Miro depletion during PINK1/Parkin-dependent mitophagy can also drive a loss of mitochondrial Myo19 upon mitochondrial damage. Finally, aberrant positioning of mitochondria in Miro1/2 double-knockout cells leads to disruption of correct mitochondrial segregation during mitosis. Thus, Miro proteins can fine-tune actin- and tubulin-dependent mitochondrial motility and positioning, to regulate key cellular functions such as cell proliferation.

**Keywords** micropattern; myosin XIX; Rhot1; Rhot2; mitofusin
**Subject Categories** Cell Adhesion, Polarity & Cytoskeleton; Membrane & Intracellular Transport
**The EMBO Journal (2018) 37: 321–336**

## Introduction

Mitochondria are critical for ATP provision and play other essential roles in cells such as buffering calcium and lipid synthesis (MacAskill & Kittler, 2010; Sheng & Cai, 2012; Mishra & Chan, 2014). The tight regulation of mitochondrial transport and distribution is therefore crucial as it enables mitochondria to be delivered and localised to areas where they are needed. In order for mitochondria to move around the cell, they need to be coupled to motor proteins. Long-range mitochondrial transport is primarily mediated by the coupling of mitochondria to microtubule motors (kinesins and dynein), whereas the actin cytoskeleton and its associated myosin motors, notably Myosin-19 (Myo19), can mediate shorter-range mitochondrial movement and actin-dependent mitochondrial anchoring (Morris & Hollenbeck, 1995; Chada & Hollenbeck, 2003; Hirokawa & Takemura, 2005; Quintero *et al*, 2009). However, the regulatory overlap between the pathways of microtubule- and actin-dependent mitochondrial trafficking and positioning and its impact on key cellular functions remain poorly understood.

The outer mitochondrial membrane (OMM) Miro (mitochondrial Rho) GTPases and the TRAK motor adaptors have emerged as key regulators of mitochondrial trafficking and distribution by coupling mitochondria to the kinesin- and dynein-dependent microtubule transport pathways (Stowers *et al*, 2002; Fransson *et al*, 2006; Birsa *et al*, 2013; van Spronsen *et al*, 2013). Miro proteins have a C-terminal transmembrane domain for OMM targeting and two GTPase domains flanking two $Ca^{2+}$-sensing EF-hand domains (Birsa *et al*, 2013; Devine *et al*, 2016). The prevailing model proposes that Miro proteins regulate trafficking by acting as the essential receptors for mitochondrial recruitment of the TRAK adaptors to drive kinesin- and dynein-mediated movements (MacAskill & Kittler, 2010; Saxton & Hollenbeck, 2012; Schwarz, 2013; Maeder *et al*, 2014; Mishra & Chan, 2014; Sheng, 2014). Miro proteins are also important targets of Parkinson's disease associated mitophagy pathway, driven by the kinase PINK1 (PTEN-induced putative kinase 1) and the ubiquitin ligase Parkin, which work together to degrade damaged mitochondria (Youle & Narendra, 2011; Covill-Cooke *et al*, 2017). Upon mitochondrial damage, Miro is rapidly ubiquitinated and depleted to block the microtubule-dependent transport of damaged mitochondria (Wang *et al*, 2011; Birsa *et al*, 2014). In mammals, two Miro family members exist, Miro1 and Miro2, with 60% sequence similarity, but little is known regarding their specific roles in regulating mitochondrial dynamics. Moreover, whether Miro proteins are

Department of Neuroscience, Physiology and Pharmacology, University College London, London, UK
*Corresponding author. Tel: +44 (0) 2076793218; E-mail: j.kittler@ucl.ac.uk

additionally involved in coordinating myosin motors and actin-dependent positioning of healthy or damaged mitochondria remains unclear.

Correct mitochondrial positioning within cells has emerged as critical for many key cellular processes including cell division, migration, signalling and survival (Youle & van der Bliek, 2012; Mishra & Chan, 2014; Morlino *et al*, 2014). The symmetric partitioning of mitochondria through both actin- and microtubule-dependent processes has recently been shown to be important for cell division (Rohn *et al*, 2014; Chung *et al*, 2016). Through the microtubule binding protein CENPF, Miro1 can promote mitochondrial redistribution following cell division (Kanfer *et al*, 2015). However, the role of Miro proteins for symmetric partitioning of mitochondria to daughter cells remains unclear.

Here, we use mouse knockout (KO) approaches to generate Miro KO embryos and mouse embryonic fibroblasts (MEFs) for Miro1, Miro2 or both proteins, allowing a detailed characterisation of their roles in regulating mitochondrial trafficking and motor adaptor recruitment. Using micropatterned substrates to normalise cell size, we dissect the different roles of Miro1 and Miro2 in mediating mitochondrial distribution. Unexpectedly, we find TRAK proteins can still localise to mitochondria in the complete absence of Miro, while Myo19 is critically dependent on Miro for its stability on the OMM. In addition, loss of both Miro proteins in Miro double-knockout (Miro$^{DKO}$) cells leads to defects in mitosis and mitochondrial segregation to daughter cells. Our work supports a revised model for Miro function in regulating both microtubule- and actin-dependent mitochondrial positioning to regulate key cellular functions.

# Results

## Differential requirements for Miro1 and Miro2 during embryonic development

We recently showed that Miro1 knockout (Miro1$^{KO}$) animals die perinatally while Miro2 knockout (Miro2$^{KO}$) animals were found to develop normally (Fig EV1A) and be viable until adulthood (Lopez-Domenech *et al*, 2016). Due to the high homology between Miro1 and Miro2 (Fransson *et al*, 2003), it is conceivable that both proteins show some degree of compensation, and thus, we wanted to investigate the consequences of deleting both Miro proteins on embryonic development. To this end, we crossed animals that were heterozygous for both genes (Miro1$^{+/-}$; Miro2$^{+/-}$ × Miro1$^{+/-}$; Miro2$^{+/-}$) and analysed the litters at different stages (Table EV1). We observed that embryos harbouring only one copy of Miro2 (Miro1$^{KO}$/Miro2$^{het}$) were present until P0 but were not viable beyond this stage (Table EV1), like Miro1$^{KO}$ animals (Nguyen *et al*, 2014; Lopez-Domenech *et al*, 2016). In contrast, embryos with only one allele of Miro1 (Miro1$^{het}$/Miro2$^{KO}$) were only found to be viable until E12.5, indicating that only one copy of Miro1 is not enough to compensate the lack of Miro2 beyond E12.5 (Table EV1 and Fig 1A–C). Importantly, Miro$^{DKO}$ embryos were only found up to embryonic stage 10.5 (E10.5) and presented reduced size and developmental defects such as uncompleted neural tube closure (Fig 1D). Interestingly, yolk sac capillaries were absent, suggesting that the development of Miro$^{DKO}$ embryos stopped at a stage prior to vascularisation

(Fig 1E). Indeed, Miro$^{DKO}$ embryos at an earlier stage (E8.5) were indistinguishable from their littermates (Table EV1).

Thus, Miro1 function seems to be critical in late development, probably allowing the inflation of the lungs in neonates, a function that cannot be compensated by Miro2 (Nguyen *et al*, 2014; Lopez-Domenech *et al*, 2016). Conversely, two copies of Miro1 are necessary to overcome early stages of development by compensating a function of Miro2 that seems critical at early stages, around E12.5 (Fig 1A–C). Consistent with this view, we observed an increase in Miro1 protein levels in Miro2$^{KO}$ embryos at E10.5, suggesting that high levels of Miro1 protein may compensate the lack of Miro2 at this stage (Figs 1F and EV1B). Interestingly, no compensatory mechanisms seem to be in place at a later time point (E12.5) where Miro1 and Miro2 protein levels closely correlate with genetic dose (Figs 1G and EV1C).

## Miro1 and Miro2 cooperate to regulate key aspects of mitochondrial morphology and distribution

To study the specific roles of Miro1 and Miro2 for mitochondrial morphology and distribution, we generated mouse embryonic fibroblast (MEF) cell lines from E8.5 embryos of all the different genetic outcomes of Miro1$^{+/-}$; Miro2$^{+/-}$ × Miro1$^{+/-}$; Miro2$^{+/-}$ crosses. The genotype of the different cell lines (confirmed by PCR amplification; Fig EV2A) correlated with the protein levels of Miro1 and Miro2 (Fig EV2B). No major change in protein content was observed across these cell lines in Western blots against actin, β-tubulin, the mitochondrial markers Tom20 or COX IV or the endoplasmic reticulum (ER) marker protein disulphide isomerase (PDI; Fig EV2B). To determine the impact on mitochondrial morphology of Miro1, Miro2 or Miro1/2 deletion, we imaged MitoTracker-labelled mitochondria in the different cell lines. We observed that Miro1$^{KO}$ and Miro2$^{KO}$ cells showed indistinguishable mitochondrial morphologies from those found in WT cells, whereas Miro$^{DKO}$ cells showed an increase in the fraction of cells with short and rounded mitochondria and a decrease in the fraction of cells with long, tubular and interconnected mitochondria (Fig 2A and B). Despite an impact on mitochondrial morphology, the maximal respiratory capacity of the electron transfer system (ETS), the normalised respiration flux (R/E) and the maximum capacity of complex IV (C-IV) were not significantly different among all genotypes either using glucose as a substrate (Fig EV2C) or with a non-glycolytic substrate (Fig EV2D), suggesting that Miro proteins are not critical in regulating respiration rate or overall energetic metabolic state of the cell.

We noted that mitochondria in Miro1$^{KO}$ cells were accumulated near the nucleus and seemed unable to reach distal regions when compared to their WT controls in accordance with previous reports (Nguyen *et al*, 2014), an effect that was greatly accentuated in Miro$^{DKO}$ cells. To determine mitochondrial distribution in the different MEF lines with high accuracy, we controlled cell size and shape using printed micropattern adhesive cell substrates (see Materials and Methods; Fig EV3A). This allows quantification of cellular parameters over many cells with an identical size and shape, greatly reducing the large inherent variability of MEF cell morphology and, hence, mitochondrial distribution (Chevrollier *et al*, 2012; Fig 2C). To measure distribution of the mitochondrial network, we performed a Sholl-based analysis of mitochondrial signal (Lopez-Domenech *et al*, 2016; Fig EV3B and C) and plotted the cumulative

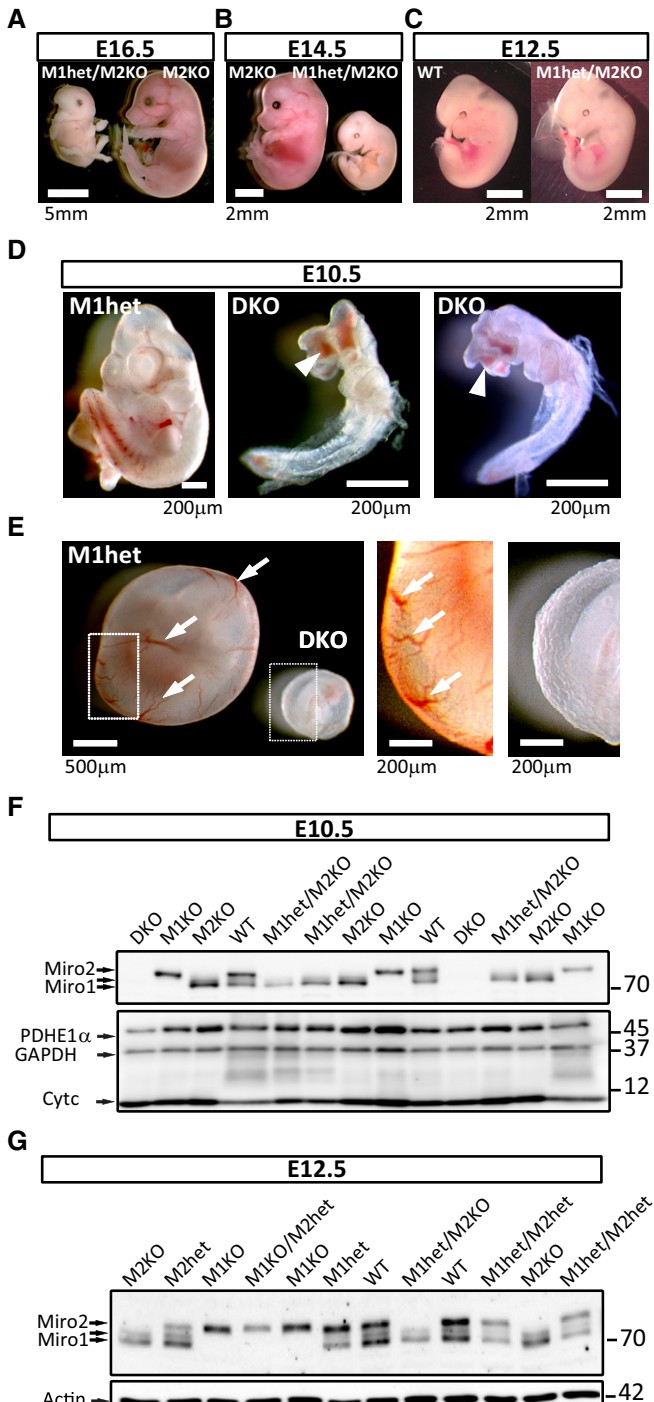

**Figure 1. Miro1 and Miro2 function is critical during early embryonic development.**

A–C  Miro1[het]/Miro2[KO] embryos (with only one copy of Miro1) are not viable beyond day E12.5 of gestation as opposed to Miro2[KO] embryos (with two copies of Miro1). At E16.5 (A) and E14.5 (B), all embryos that were identified *post hoc* as heterozygotes for Miro1 and knock out for Miro2 were found in advanced state of reabsorption. At E12.5 (C), half of the embryos of this genotype were found to be indistinguishable from WT control animals. A viable embryo was selected as a control animal for comparison. See also Table EV1.

D, E  Miro[DKO] embryos were found to be not viable from E10.5. (D) At this stage, they were very small and presented malformations and oedema in head and viscera compared with viable littermates. Neural tube closure was incomplete (arrowheads). (E) Further observation showed that Miro[DKO] embryos at E10.5 failed in generating the vasculature that irrigates the yolk sac (arrows).

F  Western blot analysis of E10.5 heads (or whole body for Miro[DKO] embryos) showing the specificity of the different bands recognised by the antibody (anti-Miro1 from Atlas) and the complete depletion of Miro1 and Miro2 proteins in Miro[DKO] embryos.

G  Western blot analysis of brains from E12.5 embryos showing that the protein levels correlate with the genetic dosage of Miro1 and Miro2. Quantification of Miro1 and Miro2 protein levels provided in Fig EV1.

Source data are available online for this figure.

the MPM and a decreased Mito[95] value, indicating that mitochondria were significantly more concentrated in proximal regions of the cell compared to either WT or Miro2[KO] cells (Fig 2C–E). Interestingly, Miro[DKO] cells showed a substantially more accentuated perinuclear accumulation of mitochondria when compared to Miro1[KO] cells, indicating that Miro2 also plays an important non-redundant role in regulating mitochondrial distribution [Fig 2C–E; Mito[95]: WT $22.16 \pm 0.20$, Miro1[KO] $19.54 \pm 0.43$, Miro2[KO] $21.43 \pm 0.26$ and Miro[DKO] $17.56 \pm 0.27$; ANOVA and *post hoc* Newman–Keuls (ANOVA-NK)]. Interestingly, MEF cell lines with only one allele of Miro1 or only one allele of Miro2 (Miro1[het]/Miro2[KO] or Miro1[KO]/Miro2[het], respectively) presented a mitochondrial distribution indistinguishable from that of Miro[DKO] cells (Figs 2D and E, and EV3C, E and F), indicating that only one copy of Miro1 or Miro2 is not sufficient to maintain an appropriate mitochondrial distribution in the proximo-distal axis. In contrast, the distribution of the nucleus was unaffected in Miro[DKO] cells, indicating that the altered mitochondrial distribution in the different genotypes is not due to an altered position of the nucleus (Fig EV3G). Thus, Miro1 and Miro2 work together in coordinating the overall distribution of the mitochondrial network within cells.

## Miro1 and Miro2 differentially regulate mitochondrial transport

Miro1 is a key regulator of mitochondrial trafficking in neurons (Macaskill *et al*, 2009; Wang & Schwarz, 2009; Lopez-Domenech *et al*, 2016) and other cell types (Saotome *et al*, 2008; Morlino *et al*, 2014; Stephen *et al*, 2015; Schuler *et al*, 2017). The prevailing model of Miro function is that it acts as the essential receptor for recruiting the motor/adaptor complexes to the mitochondria to drive mitochondrial transport along the microtubule tracks (MacAskill & Kittler, 2010; Saxton & Hollenbeck, 2012; Schwarz, 2013; Maeder *et al*, 2014; Sheng, 2014). However, whether Miro1 and Miro2 have overlapping or complementary functions and

distribution of mitochondrial signal as a function of distance from the cell centre or Mitochondrial Probability Map (MPM; Fig 2D). Using this approach, we could define the distance from the cell centre where different proportions of mitochondrial mass are found (Mito[50] or 50th percentile; Mito[90] or 90th percentile and Mito[95] or 95th percentile) across the different genotypes (Figs 2E and EV3D–F). Using the Mito[95] value, which showed more accuracy at describing differences in mitochondrial distribution, we observed that mitochondria in the Miro1[KO] cell lines showed a clear shift to the left in

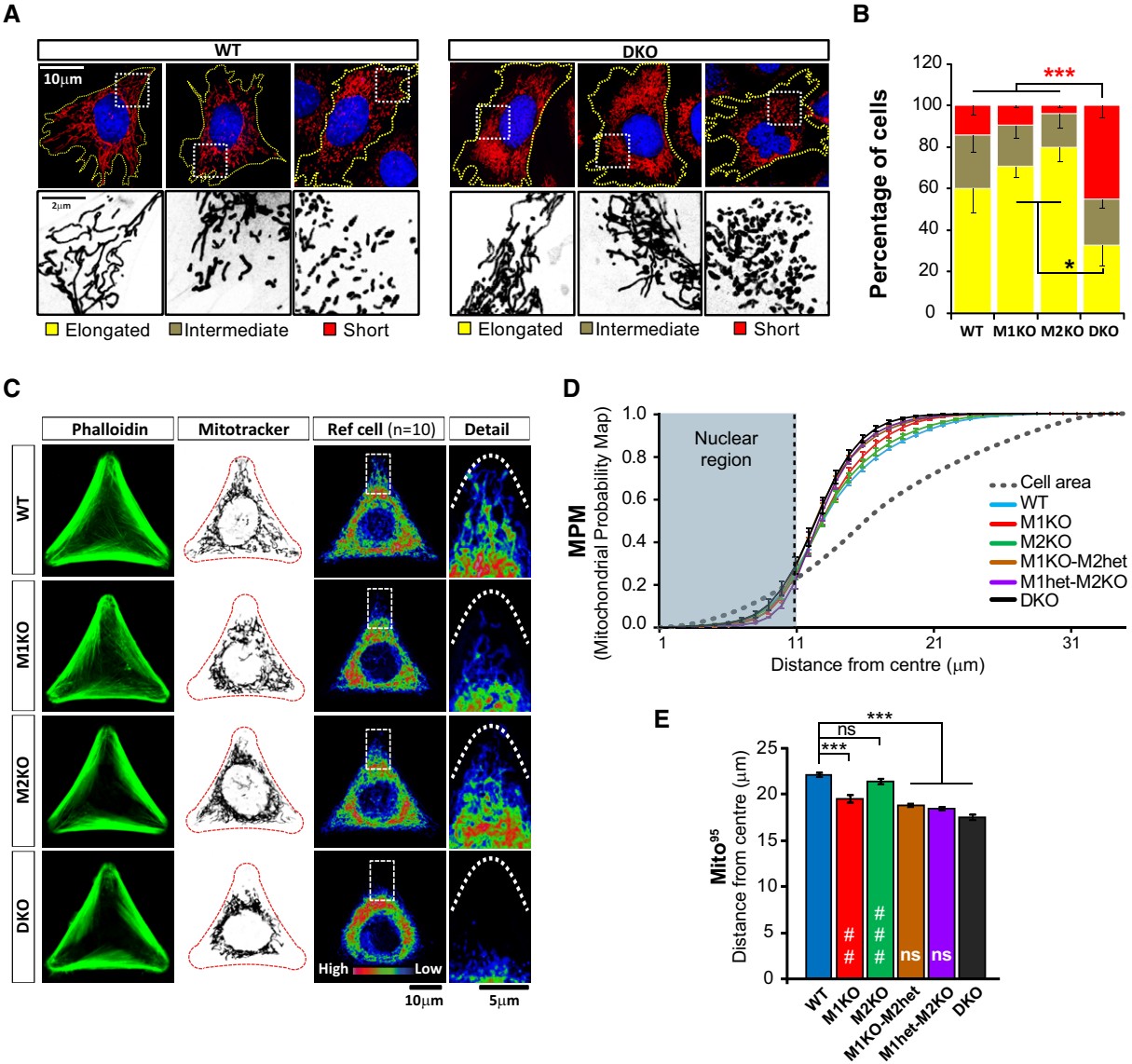

**Figure 2. Characterisation of mitochondrial morphology and distribution in mouse embryonic fibroblasts (MEFs).**

A, B   Analysis of mitochondrial morphology in Miro knockout MEFs. Examples of mitochondrial morphology in WT and Miro[DKO] cells (A). Cells were scored depending on the morphology of the majority of their mitochondrial population as elongated, short or intermediate. Graph in (B) shows that Miro[DKO] MEF cells exhibited more often short and less elongated mitochondria. Data pooled from three independent experiments (*n* = 3; ANOVA-NK). In each experiment, mitochondrial morphology was analysed from two different MEF cell lines per genotype.

C   Confocal images showing cellular morphology (phalloidin, left panels) and mitochondrial distribution (MitoTracker, middle panels) from cells growing in "Y"-shaped micropatterns from the main Miro knockout cell lines. Mitochondrial distribution differences between cell lines are evident when constructing a reference cell (right panel and boxed detail) or "heat map" by projecting the signal from 10 cells from the same genotype.

D   The cumulative distribution of mitochondrial signal or Mitochondrial Probability Map (MPM) is plotted for the different genotypes. A displacement to the left compared to WT indicates that mitochondrial signal is accumulated towards the centre of the cell. The grey dotted line represents the theoretical distribution of a homogeneously distributed signal. Analysis was performed from at least three independent experiments (number of experiments: WT 9; Miro1[KO] 6; Miro2[KO] 6; Miro1[KO]/Miro2[het] 3; Miro1[het]/Miro2[KO] 4; Miro[DKO] 9; ANOVA-NK) where at least 20 cells were analysed per genotype and experiment. Two different cell lines were used per genotype.

E   Graph showing the calculated Mito[95] values (95[th] percentile) which represent the distance from the cell centre at which 95% of the mitochondrial signal is found. Analysis was performed from at least three independent experiments (number of experiments: WT 9; Miro1[KO] 6; Miro2[KO] 6; Miro1[KO]/Miro2[het] 3; Miro1[het]/Miro2[KO] 4; Miro[DKO] 9; ANOVA-NK) where at least 20 cells were analysed per genotype and experiment. Two different cell lines were used per genotype.

Data information: Error bars represent s.e.m. Statistical significance: *$P < 0.05$ and ***$P < 0.001$ compared to WT; ##$P < 0.01$ and ###$P < 0.001$ compared to Miro[DKO].

whether the total absence of Miro could still permit any mitochondrial transport in mammalian cells have not been addressed. To address this question, we performed time-lapse imaging

experiments in MEFs expressing mitochondrially targeted DsRed2 (MtDsRed) and determined mitochondrial displacement. Miro1[KO] and Miro2[KO] cells showed a significant reduction in mitochondrial

displacements (the percentage of mitochondria that changed their position over a 10-s period) compared to WT. This decrease was even more drastic in Miro$^{DKO}$ MEFs (Fig 3A and B, and Movie EV1; mitochondrial displacement (% of area): WT 15.7 ± 0.8; Miro1$^{KO}$ 10.7 ± 0.3; Miro2$^{KO}$ 11.8 ± 0.6 and Miro$^{DKO}$ 8.4 ± 0.3, $P < 0.05$, ANOVA-NK), demonstrating that Miro1 and Miro2 can both regulate aspects of mitochondrial trafficking. In contrast, lysosomal displacement was unaffected in Miro$^{DKO}$ MEFs compared to WT (Appendix Fig S1A and B, and Movie EV2), indicating that the transport defects are specific to mitochondrial transport.

Mitochondria in mammalian cells can move using the actin cytoskeleton for short-range displacements and the microtubule cytoskeleton for longer range movements (Morris & Hollenbeck, 1995). Surprisingly, mitochondria could still often be found aligned with microtubule filaments in Miro$^{DKO}$ cells (Appendix Fig S1C), suggesting that association of mitochondria to microtubule tracks can occur even in the complete absence of any Miro. We quantified the number of fast tubulin-dependent mitochondrial transport events in our movies, characterised as directional displacements of mitochondria that covered at least 5 μm in distance and moved faster than 0.15 μm/s. We observed that the number of micro-tubule-dependent mitochondrial trafficking events was significantly reduced in Miro1$^{KO}$ cells but was unaltered in Miro2$^{KO}$ cells

(Fig 3C). Unexpectedly, in Miro$^{DKO}$ cells, a small number of directed mitochondrial movements could still be detected (Fig 3C and D, and Movie EV3). They were found to be greatly decreased compared to WT cells but, interestingly, not different compared to Miro1$^{KO}$ cells (Fig 3C; number of runs: WT 7.63 ± 0.96, M1KO 2.75 ± 0.32, Miro2$^{KO}$ 6.92 ± 0.72 and Miro$^{DKO}$ 3.15 ± 0.27; $P < 0.05$, ANOVA-NK). Disrupting the microtubule cytoskeleton with vinblastine

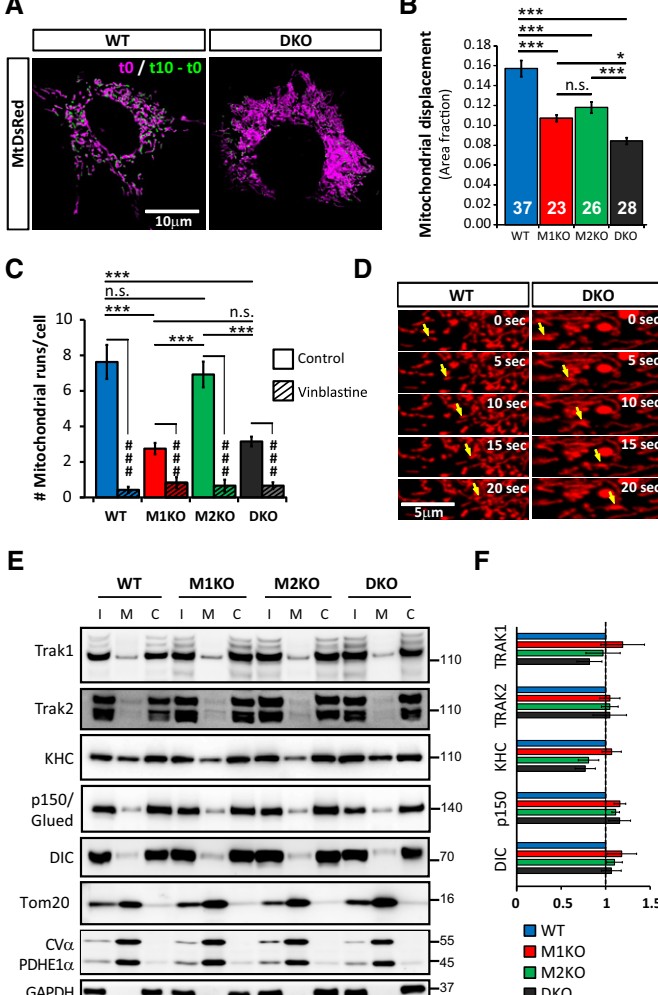

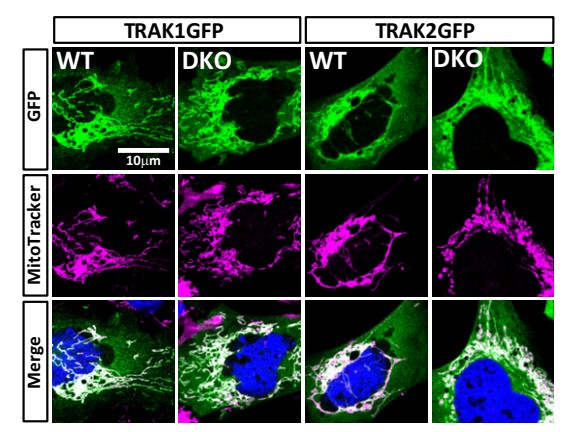

**Figure 3. Mitochondrial movement is reduced but not completely abolished in Miro$^{DKO}$ MEFs.**

A, B Mitochondrial displacement is reduced in all genotypes compared to WT. (A) Stills from movies showing the mitochondrial compartment from WT and Miro$^{DKO}$ MEFs at time = 0 (magenta) and the new area occupied by mitochondria 10 s later (mitochondrial area at $t = 10$ s − mitochondrial area at $t = 0$) (green). (B) Mitochondrial displacement at a given time point ($t_n$) was calculated by subtracting mitochondrial area at two different time points separated by 10 s and normalised to total mitochondrial area from that given time point: $(t_{n+10} - t_n)/t_n$. The final displacement value was averaged over the 59 pairs of frames for each movie (59 pairs over a 61 frame movie). Data obtained from the indicated number of cells from six different experiments ($n$ = number of cells; ANOVA-NK).

C, D Mitochondrial transport on microtubule tracks is not completely abolished in Miro$^{DKO}$ MEFs. The number of tubulin-dependent mitochondrial runs (C) was equally decreased (but not abolished) in Miro1$^{KO}$ and in Miro$^{DKO}$ cells but unaffected in Miro2$^{KO}$ cells when compared to WT ($n$ = number of cells; ANOVA-NK; data obtained from the same cells as in B). Disrupting microtubules with vinblastine abolished mitochondrial runs. Data for vinblastine treatment obtained from three independent experiments ($n$ = number of cells; WT = 14, Miro1$^{KO}$ = 6; Miro2$^{KO}$ = 6; Miro$^{DKO}$ = 14; $t$-test inside genotypes). (D) Stills from the movies quantified in (A) and (B) showing fast and directed mitochondrial movements (yellow arrows) in Miro$^{DKO}$ cells.

E–G TRAK1, TRAK2 and motor complex components are recruited to the mitochondria even in the absence of Miro. (E) Western blots showing that TRAK1, TRAK2 and kinesin heavy chain, P150/Glued and the dynein intermediate chain can be found in purified mitochondrial fractions even in Miro$^{DKO}$ MEFs (I: input; M: mitochondrial fraction; C: cytoplasmic fraction). (F) Quantification of mitochondrial enrichment (mitochondrial signal/cytoplasmic signal) of the indicated adaptor/motor components and normalised to WT. Data compiled from four independent subcellular fractionations ($n$ = number of fractionations; ANOVA-NK). (G) Confocal micrographs showing that exogenous TRAK1GFP and TRAK2GFP (green) are also enriched in mitochondria (magenta) in Miro$^{DKO}$ cells.

Data information: Error bars represent s.e.m. Statistical significance: *$P < 0.05$ and ***$P < 0.001$; ###$P < 0.001$.
Source data are available online for this figure.

(Appendix Fig S1D) inhibited all remaining directional movements, confirming that the analysed mitochondrial runs were indeed along microtubules (Fig 3C). This observation suggests that only Miro1 and not Miro2 regulates microtubule-dependent mitochondrial trafficking and, in addition, suggests that Miro proteins, in contrast to the broadly accepted model, are not obligatory mediators of microtubule-dependent mitochondrial transport.

### TRAK motor adaptors can still form functional anterograde mitochondrial transport complexes in the absence of Miro

The unexpected observation that some Miro-independent transport of mitochondria along microtubules remains in Miro[DKO] cells prompted us to further investigate whether Miro proteins are obligate adaptors for motor protein recruitment to the mitochondrial membrane. To test this, we performed mitochondrial fractionation (Frezza *et al*, 2007) studies in our different cell lines followed by Western blotting with antibodies to Kinesin-1 and dynein/dynactin motor proteins and their TRAK adaptors (Fig 3E). Surprisingly, all the microtubule motors tested, the molecular adaptor TRAK1, and to a lesser extent, TRAK2, could still be found in the mitochondrial fraction (confirmed with specific mitochondrial and cytoplasmic markers) at similar levels in all the different genotypes (Fig 3F). In agreement with this, GFP-tagged versions of both TRAK1 and TRAK2 could also be found targeted to mitochondria when expressed in Miro[DKO] cells (Fig 3G), suggesting the presence of other TRAK acceptors on the mitochondrial membrane in Miro[DKO] cells. Indeed, proximity ligation assays (PLA) revealed that the interaction between TRAK1 and mitofusin1 [Mfn1; a previously reported TRAK interactor (Lee *et al*, 2017; Misko *et al*, 2010)] is preserved in Miro[DKO] cells, while the interaction between TRAK2 and Mfn1 appeared enhanced (Fig EV4). Thus, recruitment of TRAK1 and TRAK2 to the mitochondrial outer membrane in the absence of Miro can happen through other adaptors such as the mitofusins.

To further test the functionality of the TRAK/motor complexes upon loss of Miro, we used cell micropatterning techniques and mitochondrial Sholl analysis. GFP transfection had no effect on the MPM or Mito[95] values when compared with untransfected cells for either WT or Miro[DKO] cells (Appendix Fig S2A and B), indicating that mitochondrial distribution is not altered by transfection. In contrast, when we expressed TRAK1GFP in WT MEFs, we observed a subtle mitochondrial redistribution towards the cell periphery away from the perinuclear region with a trend towards a rightward shift in the MPM compared to untransfected cells (Fig 4A, C and E), suggesting that targeting more TRAK1 to mitochondria can enhance kinesin-mediated transport. In agreement with this, while expression of the kinesin-1 motor KIF5C alone had no effect on mitochondrial distribution (Appendix Fig S2C and D), co-expression of KIF5C with TRAK1 dramatically enhanced the redistribution of mitochondria to the cell periphery in a similar way as previously observed for TRAK2 and KIF5C in COS-7 cells (Smith *et al*, 2006), leading to a shift to the right in the MPM and a significant increase in the Mito[95] value (Fig 4A, C and E). Surprisingly, when we performed the same analysis in the absence of Miro proteins, we found that expression of TRAK1GFP in Miro[DKO] cells could still induce a significant redistribution of mitochondria towards the most distal regions of the cells, as observed by a shift to the right in the MPM and a significant increase in the Mito[95] value. This effect was further enhanced upon

co-expression of TRAK1 (or TRAK2) and KIF5C (Fig 4B and F–H) while KIF5C expression alone had no effect (Appendix Fig S2C and D), confirming that expression of TRAK1/2 can recruit more KIF to the mitochondria, even in the absence of Miro proteins.

Thus, in the absence of both Miro proteins, TRAK proteins can not only still be targeted to the mitochondrial membrane but are functional in terms of driving kinesin-mediated anterograde mitochondrial transport to the cell periphery.

### Miro1 facilitates the TRAK2-dependent retrograde redistribution of mitochondria

In contrast to what we observed for TRAK1, expression of TRAK2 alone in WT MEFs led to the accumulation of mitochondria in the perinuclear region, shown by a leftward shift in the MPM and a significant decrease in the Mito[95] value (Fig 4A, D and E). This suggests that TRAK2 expression may favour dynein-directed retrograde mitochondrial transport as previously proposed (van Spronsen *et al*, 2013). Unlike in WT cells, TRAK2 expression in Miro[DKO] cells could not drive significant retrograde redistribution of mitochondria (Fig 4B, G and H). This could be because TRAK2-dependent retrograde mitochondrial transport is occluded in Miro[DKO] MEFs, where mitochondria are already significantly clustered perinuclearly, or because Miro proteins are permissive for dynein-dependent regulation of mitochondrial distribution, as it has been previously suggested in fly neurons (Babic *et al*, 2015; Melkov *et al*, 2016). In agreement with the latter hypothesis, the perinuclear clustering of mitochondria observed in Miro[DKO] cells is reminiscent of mitochondrial distribution upon dynein inhibition (Varadi *et al*, 2004). In addition, we noted that when expressed with KIF5C, TRAK2 was very effective at driving anterograde mitochondrial transport in Miro[DKO] MEFs but not in WT MEFs (Fig 4D, E, G and H), suggesting that the absence of Miro proteins may preferentially favour kinesin-dependent anterograde mitochondrial transport by TRAK2, perhaps because dynein activity is not available to oppose this.

To reveal potential Miro1- and Miro2-specific roles in regulating motor-dependent mitochondrial positioning, we performed a similar series of experiments in Miro1 or Miro2 knockout MEF lines. TRAK1 overexpression in either Miro1[KO] or Miro2[KO] cells could drive mitochondria to the cell periphery, an effect that was greatly enhanced upon co-expression of KIF5C (Fig 5A–C, E, F and H), similar to our observations in Miro[DKO] cells. In contrast, overexpression of TRAK2 in cells depleted of Miro1 no longer induced the perinuclear redistribution of mitochondria seen in either WT or Miro2[KO] cells (Fig 5D, E, G and H) but rather had the opposite effect, moving mitochondria to the periphery. This suggests that Miro1 (but not Miro2) may facilitate TRAK2-dependent retrograde redistribution of mitochondria.

### Miro proteins recruit and stabilise Myo19 on the mitochondria

The fact that compared to Miro1[KO], Miro[DKO] cells showed an enhanced disruption of mitochondrial displacement but not tubulin-mediated long-range directional movements suggests that other transport mechanisms may also be altered upon loss of both Miro proteins. The actin cytoskeleton serves as a substrate to allow mitochondrial movement, docking and distribution (Morris & Hollenbeck, 1995; Chada & Hollenbeck, 2004). Several myosins

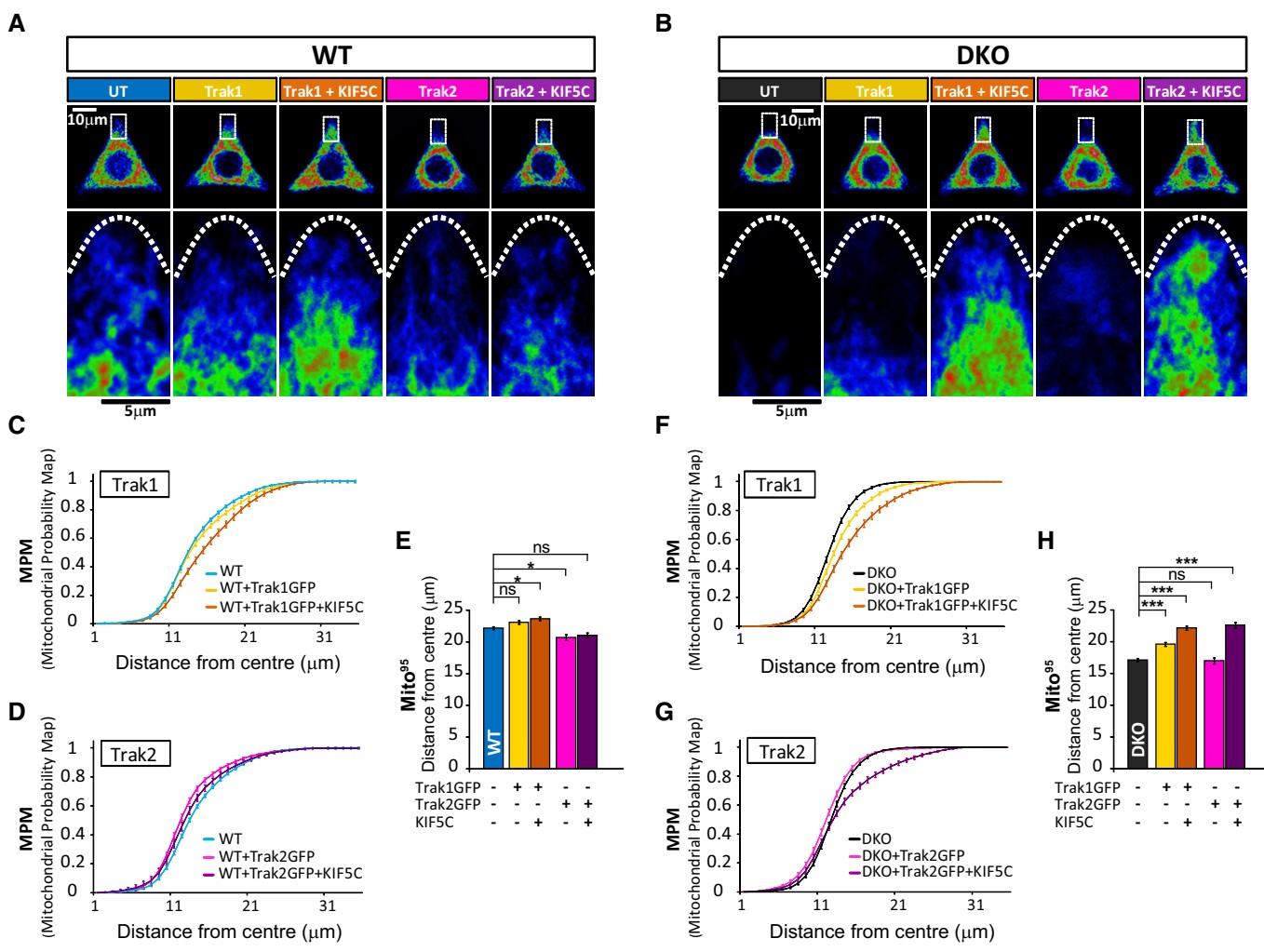

**Figure 4.  Miro proteins are not required for TRAK/kinesin-dependent anterograde movement.**

A–H   Effect of TRAK1 and TRAK2 with and without KIF5C overexpression on mitochondrial redistribution in WT (A, C, D, and E) or Miro[DKO] (B, F, G and H) cell lines. Reference cells generated by projection of 10 WT (A) or Miro[DKO] (B) cells with the same transfection combination. An inset in each cell is shown magnified below to better show the occupancy of mitochondria in the tips of the triangular cell. MPM from WT cells (C and D) or Miro[DKO] cells (F and G) overexpressing TRAK1GFP (C and F) or TRAK2GFP (D and G), respectively. (E and H) Mito[95] values calculated from the above experimental conditions in WT (E) and Miro[DKO] (H) cell lines. Data obtained from three independent experiments (*n* = number of cells; in WT: control 56; TRAK1GFP: 55; TRAK1 + KIF5C 55; TRAK2GFP 48; TRAK2GFP + KIF5C 46; and in Miro[DKO]: control 60; TRAK1GFP 61; TRAK1 + KIF5C 56; TRAK2GFP 62; TRAK2GFP + KIF5C 54; ANOVA-NK). Error bars represent s.e.m. Statistical significance: **P* < 0.05 and ****P* < 0.001.

have been related to the regulation of mitochondrial transport (Pathak *et al*, 2010) although to date only Myo19 has been shown to localise on mitochondria (Quintero *et al*, 2009; Shneyer *et al*, 2016). Interestingly, endogenous Myo19 levels were reduced in Miro2[KO] cell lines while in Miro1[KO] they remained unchanged (Fig 6A). This decrease was accentuated in Miro1[KO]/Miro2[het] and even more in Miro1[het]/Miro2[KO] cells, reaching the strongest effect in Miro[DKO] cells, where Myo19 levels were ~10% of control levels in WT cells (Fig 6A and B). This indicates that both Miro1 and Miro2 work together to maintain the endogenous levels of Myo19 and also suggests that Miro2 may have a more important role. This decrease prompted us to investigate whether the mitochondrial targeting of this myosin motor would be affected by Miro deletion. Fractionation of WT MEFs revealed that endogenous Myo19 is heavily enriched

on mitochondria with an additional pool of Myo19 localising in the cytoplasmic fraction (Fig 6C and D), suggesting that Myo19 may be able to translocate on and off the mitochondria. Loss of either Miro1 or Miro2 led to a mild decrease in the enrichment of Myo19 in the mitochondria versus the cytoplasm while loss of both Miro proteins greatly enhanced this effect with an almost complete loss of Myo19 from mitochondria and the remaining Myo19 found exclusively in the cytosol in Miro[DKO] cells (Fig 6C and D). Similar results were obtained by confocal microscopy, which revealed that compared to WT cells, the pool of Myo19 co-localising with Mitotracker-labelled mitochondria was decreased in Miro1[KO] and Miro2[KO] cells and, to a larger extent, in Miro[DKO] cells (Fig 6E and F). Importantly, overexpression of either Miro1GFP or Miro2GFP could readily rescue the mitochondrial pool of endogenous Myo19 in Miro[DKO] cells, further

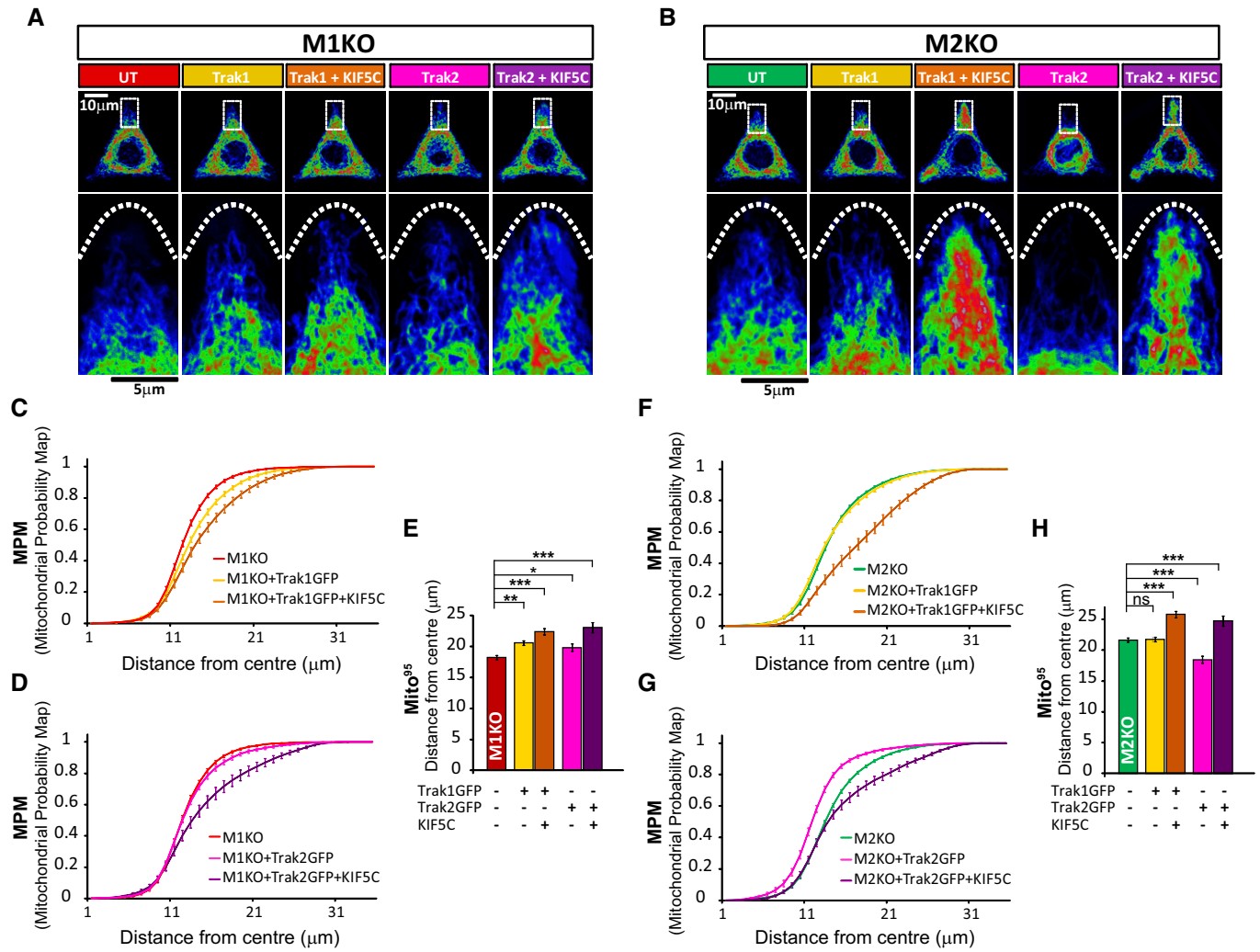

**Figure 5.  TRAK2 has a preference for Miro1 in regulating retrograde mitochondrial trafficking.**

A–G    Effect of TRAK1 and TRAK2 with and without KIF5C overexpression on mitochondrial redistribution in Miro1[KO] (A, C–E) or Miro2[KO] (B, F–H) cell lines. MPM from Miro1[KO] cells (C and D) or Miro2[KO] (F and G) overexpressing TRAK1GFP (C and F) or TRAK2GFP (D and G), respectively. (E and H) Mito[95] values calculated from the above experimental conditions in Miro1[KO] (E) and Miro2[KO] (H) cell lines. Data obtained from three independent experiments (*n* = number of cells; in Miro1[KO]: control 55; TRAK1GFP: 49; TRAK1 + KIF5C 33; TRAK2GFP 40; TRAK2GFP + KIF5C 41; and in Miro2[KO]: control 55; TRAK1GFP 51; TRAK1 + KIF5C 35; TRAK2GFP 64; TRAK2GFP + KIF5C 38; ANOVA-NK). Error bars represent s.e.m. Statistical significance: *$P < 0.05$, **$P < 0.01$ and ***$P < 0.001$.

confirming the specificity of the stabilisation of mitochondrial Myo19 levels (Fig EV5A and B). In light of the reduced levels of Myo19 in the mitochondrial fraction, we hypothesised that Miro proteins could act as receptors of Myo19 to the mitochondria, thereby protecting it from cytoplasmic degradation.

To test this hypothesis, we overexpressed a GFP-tagged version of Myo19 (GFPMyo19) in our MEF lines. GFPMyo19 localised predominantly in the mitochondrial compartment in WT cells with only a small amount localising in the cytoplasm (Fig 6G). This local-isation was altered in Miro1[KO] and Miro2[KO] cells which showed a clear increase in the cytoplasmic fraction of GFPMyo19 which was further enhanced in Miro[DKO] cells (Fig 6G). To analyse this balance, we calculated the ratio of mitochondrial versus cytoplasmic signal. As expected, the mitochondrial enrichment of GFPMyo19 localisa-tion in WT cells was significantly decreased in Miro1[KO] and Miro2[KO] cell lines and even further in Miro[DKO] cells (Fig 6H), supporting that

both Miro proteins are implicated in the mitochondrial localisation of Myo19. As expected, co-expression of either Miro1myc or Miro2myc completely rescued the mitochondrial enrichment of GFPMyo19 in Miro[DKO] cells (Fig 6I and J, and EV5C). In contrast, Miro1 and Miro2 lacking the mitochondrial targeting signal (Miro1ΔTMmyc and Miro2ΔTMmyc) and which therefore localise in the cytoplasm (Fransson *et al*, 2006) were not able to recruit GFPMyo19 to the mitochondria of Miro[DKO] cells (Fig 6I and J, and EV5C). Instead, cytoplasmic Miro1ΔTMmyc and Miro2ΔTMmyc, when co-expressed in WT cells, could recruit GFPMyo19 to the cyto-plasm altering its mitochondrial localisation to levels similar to Miro[DKO] cells (Fig 6K and L, and EV5C). These results strongly support a role for Miro1 and Miro2 as mitochondrial receptors with the ability to recruit Myo19 to the mitochondria. In agreement with this, we found that when co-expressed in COS-7 cells, GFPMyo19 could readily co-precipitate either Miro1myc or Miro2myc,

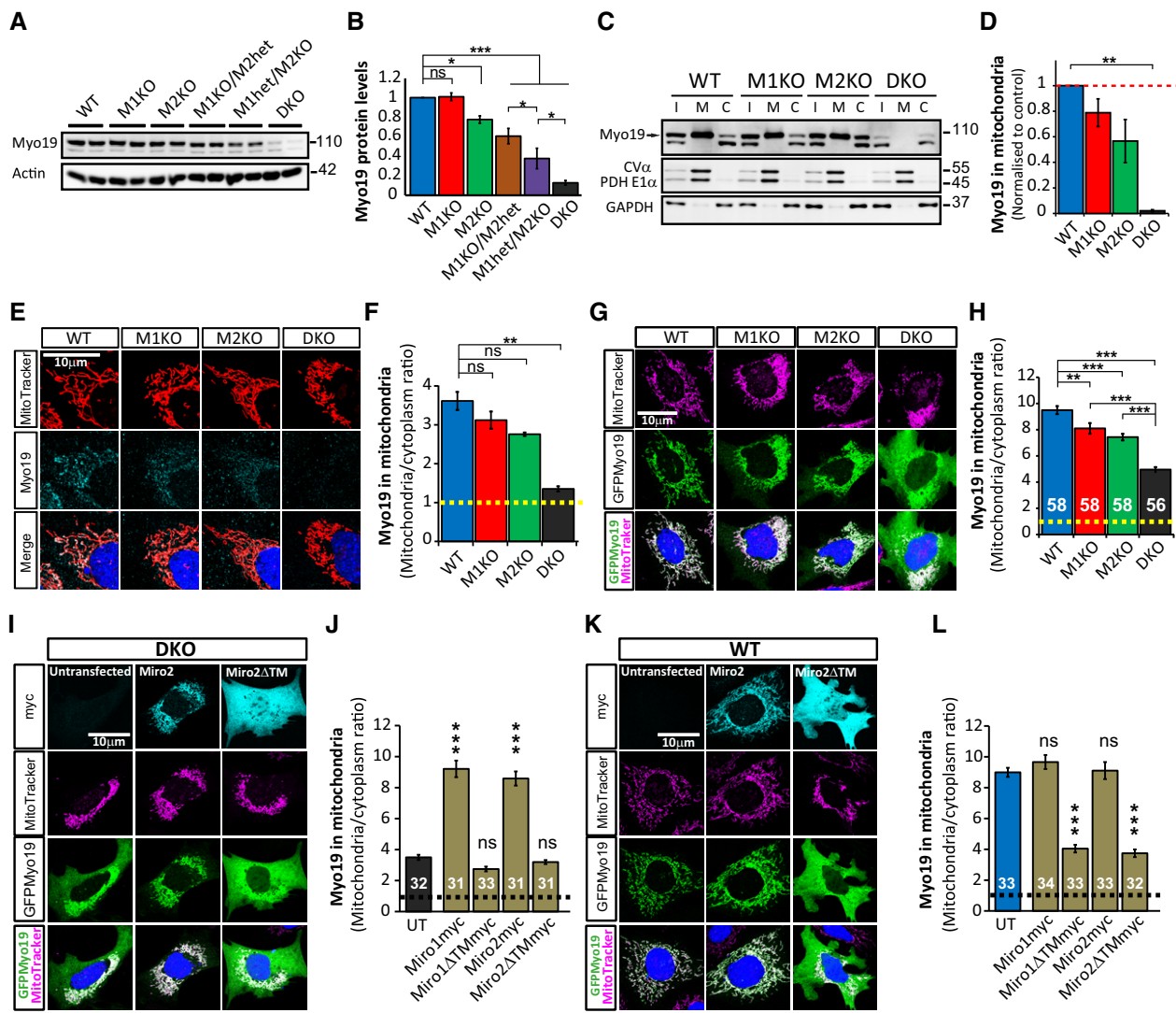

**Figure 6. Miro proteins recruit Myo19 to the mitochondria.**

A, B   Myo19 levels correlate with the quantity of Miro proteins in mouse fibroblasts. Western blot (A) and quantification (B) of Myo19 protein levels in all different genotypes.

C, D   Myo19 enrichment in the mitochondria is dependent on Miro proteins. (C) Western blot of endogenous Myo19 in the different cellular fractions from WT, Miro1[KO], Miro2[KO] and Miro[DKO] MEFs. I: input; M: mitochondrial fraction; C: cytoplasmic fraction. (D) Enrichment of Myo19 on mitochondria calculated as mitochondrial signal/cytoplasmic signal and normalised to WT ($n = 5$ independent experiments; Kruskal–Wallis test with Dunn's correction).

E   Representative images of endogenous immunostained Myo19 and Mitotracker Orange in MEFs.

F   Enrichment of Myo19 in mitochondria (ratio of mitochondrial Myo19 signal to non-mitochondrial signal) and comparison between WT (70 cells), Miro1[KO] (67 cells), Miro2[KO] (64 cells) and Miro[DKO] (54 cells) MEFs ($n = 3$ independent experiments; ANOVA-NK).

G   Representative confocal images of cellular localisation of expressed GFPMyo19.

H   Quantified mitochondrial enrichment of GFPMyo19 signal (ratio of mitochondrial Myo19 signal to non-mitochondrial signal) in the different MEF cell lines. Data come from the indicated number of cells from three different experiments ($n$ = number of cells; ANOVA-NK).

I–L   Miro proteins recruit Myo19 to the mitochondria. Miro[DKO] cells (I) and WT (K) expressing GFPMyo19 alone or together with the mitochondrial Miro2myc or the cytosolic Miro2ΔTMmyc (see Fig EV5C for the equivalent experiment with Miro1myc and Miro1ΔTMmyc). Quantification of mitochondrial enrichment of GFPMyo19 (ratio of mitochondrial signal to non-mitochondrial signal) in the indicated conditions in Miro[DKO] (J) and WT (L). Data obtained from the indicated number of cells from three different experiments ($n$ = number of cells; ANOVA-NK). UT, untransfected.

Data information: Error bars represent s.e.m. Statistical significance: *$P < 0.05$, **$P < 0.01$ and ***$P < 0.001$.
Source data are available online for this figure.

supporting their ability to form protein complexes on the mitochondrion (Fig 7A).

The dramatic loss of mitochondrial Myo19 upon genetic deletion of Miro suggested that Myo19 may be a highly labile protein and

that Miro proteins play a critical role in Myo19 stabilisation at the mitochondria. To test this hypothesis, we performed time-course experiments using cycloheximide, an inhibitor of protein synthesis, to test the stability of Myo19 in the presence or absence of the

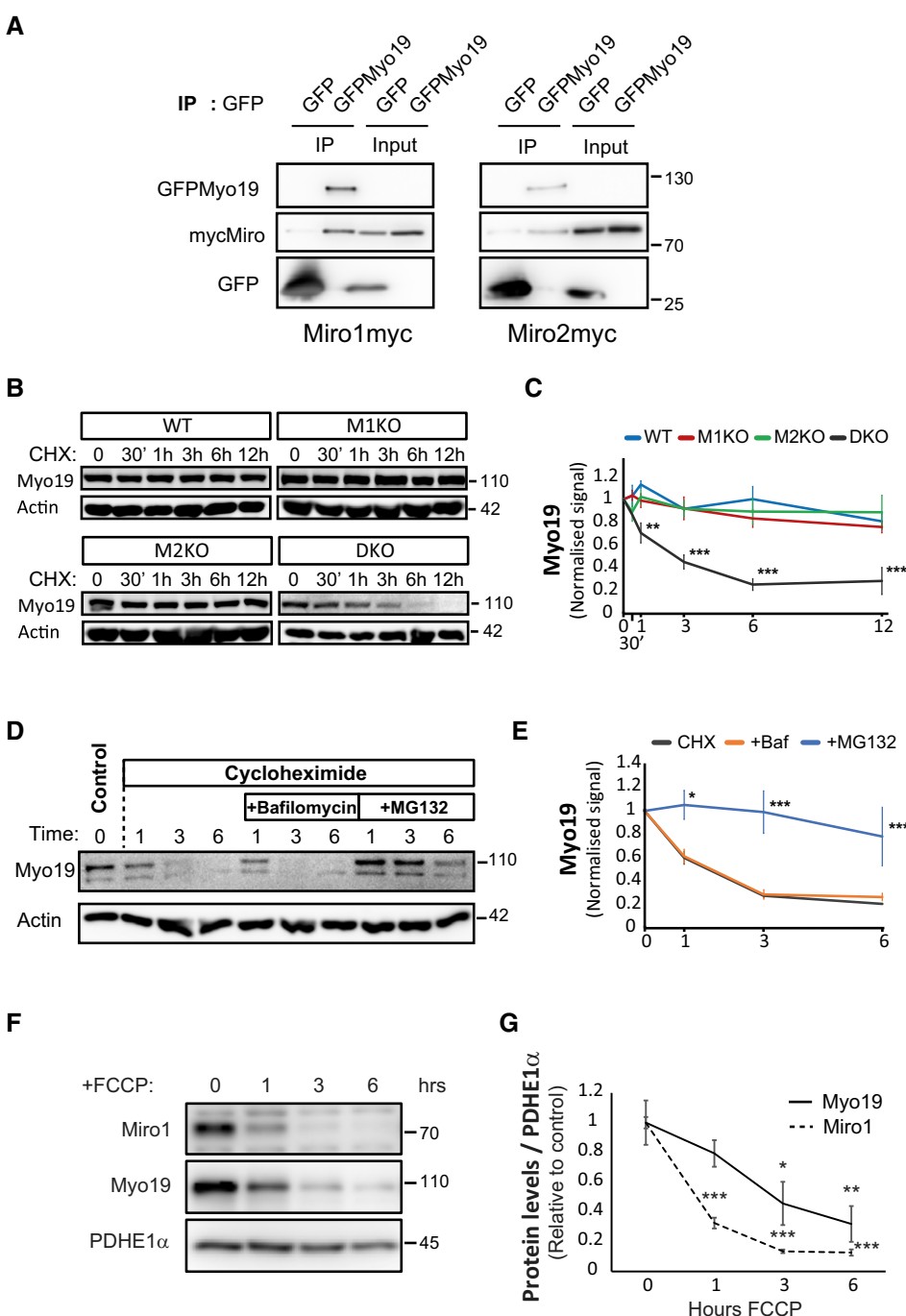

**Figure 7. Miro proteins stabilise Myo19 on the mitochondria.**

A    Co-immunoprecipitation showing that both Miro1myc and Miro2myc interact with GFPMyo19 (Western blot representative of three different experiments).

B–E  Endogenous Myo19 is stabilised on the mitochondria by Miro proteins. (B) Representative Westerns blots of time-course treatments of 1 μM cycloheximide for the indicated times in the different MEF knockout cell lines. (C) Graphical representation of the normalised levels of Myo19 showing that when protein synthesis is inhibited, the levels of Myo19 in Miro[DKO] cells disappear in ~6 h while they remain stable in the rest of the genotypes (n = 3 independent experiments; ANOVA-NK in each time point). Representative Western blot (D) of Miro[DKO] cells treated at the indicated times with 1 μM cycloheximide and the combination of 1 μM cycloheximide and 400 nM bafilomycin or 10 μM of MG132 to block lysosomal-dependent protein degradation and proteosomal degradation, respectively. The normalised levels of Myo19 are plotted in (E) and show that Myo19 degradation is dependent on the proteasome (n = 3 independent experiments; ANOVA-NK in each time point).

F, G  Representative Western blot (F) and quantification (G) showing that Myo19 is readily degraded after FCCP (10 μM) treatment closely following Miro1 degradation (n = 3 independent experiments; ANOVA-NK in each condition compared to time = 0).

Data information: Error bars represent s.e.m. Statistical significance: *P < 0.05, **P < 0.01 and ***P < 0.001.
Source data are available online for this figure.

different Miro proteins. Endogenous Myo19 levels remained stable up to 12 h of cycloheximide treatment in WT cells and showed similar stability in Miro1$^{KO}$ and Miro2$^{KO}$ cell lines (Fig 7B and C). In contrast, Myo19 levels in Miro$^{DKO}$ cells rapidly dropped under cycloheximide treatment (Fig 7B and C) due to a degradation process dependent on the proteasome and independent of the lysosomal degradation pathway (Fig 7D and E). These results indicate that the cytoplasmic pool of Myo19 is prone to degradation and suggest that when localised on the mitochondria, Myo19 is protected from degradation and thus stabilised. Furthermore, we tested the impact on Myo19 levels upon rapid Parkin-mediated Miro depletion upon mitochondrial damage. Using a SH-SY5Y neuroblastoma cell line stably overexpressing Parkin, mitochondrial damage induced by the mitochondrial uncoupler FCCP (Birsa *et al*, 2014) led to a dramatic decrease of Miro1 within 1 h of treatment and almost complete loss at later time points as previously reported (Wang *et al*, 2011; Birsa *et al*, 2014). Interestingly, we also noted a dramatic loss of Myo19 levels at the later time points of 3 and 6 h of FCCP treatment closely following the loss of Miro, although with apparently slower kinetics (Fig 7F and G).

Our results support a critical role for Miro proteins as key adaptors for Myo19 mitochondrial recruitment and stabilisation to regulate Myo19 levels on the mitochondrion and moreover, demonstrate that Myo19 levels can be dynamically regulated along with Miro during mitochondrial damage.

## Loss of Miro leads to asymmetric segregation of mitochondria during mitosis and reduced mitosis rate

Myo19 has been proposed to regulate an equal segregation of mitochondria to daughter cells during mitosis (Rohn *et al*, 2014). The observation that Miro$^{DKO}$ cells have dramatically reduced levels of Myo19 prompted us to investigate whether mitochondrial segregation is affected by Miro deletion in our MEF lines. First, we noted that in Miro$^{DKO}$ MEFs mitochondrial content appeared to be more heterogeneous compared to wild-type control cells (Fig 8A). We compared the mitochondrial content per cell through a ratio of Tom20-stained area (mitochondria) to a GFP cell fill area which revealed that Miro$^{DKO}$ cells had a significantly higher variance in mitochondrial content in comparison with WT cells (Fig 8B; $P = 0.0100$, *F*-test). These differences in basal mitochondrial content would be in accordance with a disrupted segregation of mitochondria during mitosis. To investigate this possibility, we performed long-term live cell imaging over several hours, allowing us to follow mitochondrial content and distribution (revealed by Su9GFP overexpression) to daughter cells through a round of cell division. In dividing WT cells, we would most commonly observe a clear equal segregation of mitochondrial content following cytokinesis (Fig 8C). In contrast, in Miro$^{DKO}$ cells, mitochondrial content could often be seen to be highly variable between daughter cells with one cell receiving a much larger proportion of the mitochondrial network (Fig 8C and D, and Movie EV4). In addition, we also report an unequal mitochondrial segregation happening in Miro2$^{KO}$ MEFs undergoing mitosis, although in a considerably smaller proportion of cells compared to the Miro$^{DKO}$ MEFs (Fig 8D) which correlates with the levels of Myo19 observed in both cell lines (Fig 6A). Interestingly, we noticed that this unequal segregation of mitochondria correlated with a mitochondrial distribution in the original cells that

were radially asymmetric and highly polarised (Fig 8C and Movie EV4).

Possibly as a consequence of an unequal segregation of mitochondria, Miro$^{DKO}$ cells underwent significantly fewer mitotic events in comparison with WT cells (Fig 8E). Similarly, we noted a small but significant increase in the number of cells that detached from the substrate and died during the course of our movies (Appendix Fig S3A) although it is unclear whether these cells died as a consequence of a failed mitotic event. To test whether defects in mitochondrial segregation are due to the loss of Myo19 in Miro$^{DKO}$ cells, we repeated our long-term imaging experiments with cells overexpressing exogenous GFPMyo19. We observed that GFPMyo19 overexpression partially rescued the unequal segregation of mitochondria during mitosis in Miro$^{DKO}$ cells while having no effect on WT cells (Fig 8F). Interestingly, both the rate of mitotic events and the rate of cell death of Miro$^{DKO}$ cells were unchanged when GFPMyo19 was overexpressed (Fig 8G and Appendix Fig S3B), suggesting that Myo19 in the mitochondria is necessary but not sufficient to ensure correct segregation of mitochondria during mitosis. Altogether, our results suggest that Miro proteins are still necessary to ensure correct mitochondrial segregation, either by further regulating Myo19 activity in the mitochondria or by coordinating Myo19 function with the microtubule transport pathway through regulation of kinesin and dynein motor complexes.

Thus, Miro proteins accomplish a critical role in coordinating the intracellular distribution of mitochondria by both microtubule and actin motors. This coordination is necessary to maintain a homogeneous distribution of mitochondria within cells, which is critical to ensure an equal segregation of mitochondria during mitosis.

# Discussion

Here, using mouse knockout approaches to generate Miro KO embryos and MEFs, we report essential requirements for Miro proteins in mid-stage development of the mouse embryo and demonstrate a critical role of mammalian Miro proteins in coordinating mitochondrial distribution through both microtubule- and actin motor-dependent mechanisms. Finally, we demonstrate an important role of Miro-mediated mitochondrial positioning for regulating symmetric mitochondrial segregation during cell division.

Constitutive knockout of Miro1 results in early postnatal lethality (Nguyen *et al*, 2014; Lopez-Domenech *et al*, 2016) whereas Miro2$^{KO}$ animals are viable into adulthood and fertile (Lopez-Domenech *et al*, 2016). In contrast, the impact of compound Miro$^{DKO}$ had not been explored prior to this study. By generating embryos from Miro1/2 heterozygous crosses, we report here that complete loss of Miro1 and Miro2 leads to a failure in early embryonic development, with Miro$^{DKO}$ embryos unable to survive beyond E8.5. The earlier lethality observed in the Miro$^{DKO}$ embryos confirms that Miro1 and Miro2 have essential and non-redundant roles during key early developmental stages. Similar developmental abnormalities, resulting in death during early embryogenesis, have also been observed in mice lacking other genes encoding critical regulators of mitochondrial dynamics such as mitochondrial fusion proteins Mfn1 and Mfn2, knockout of which leads to developmental arrest at E8.5 and reabsorption by E10.5–E11.5 (Chen *et al*, 2003). Interestingly, the embryonic lethality observed at E12.5 in the Miro1$^{het}$/Miro2$^{KO}$

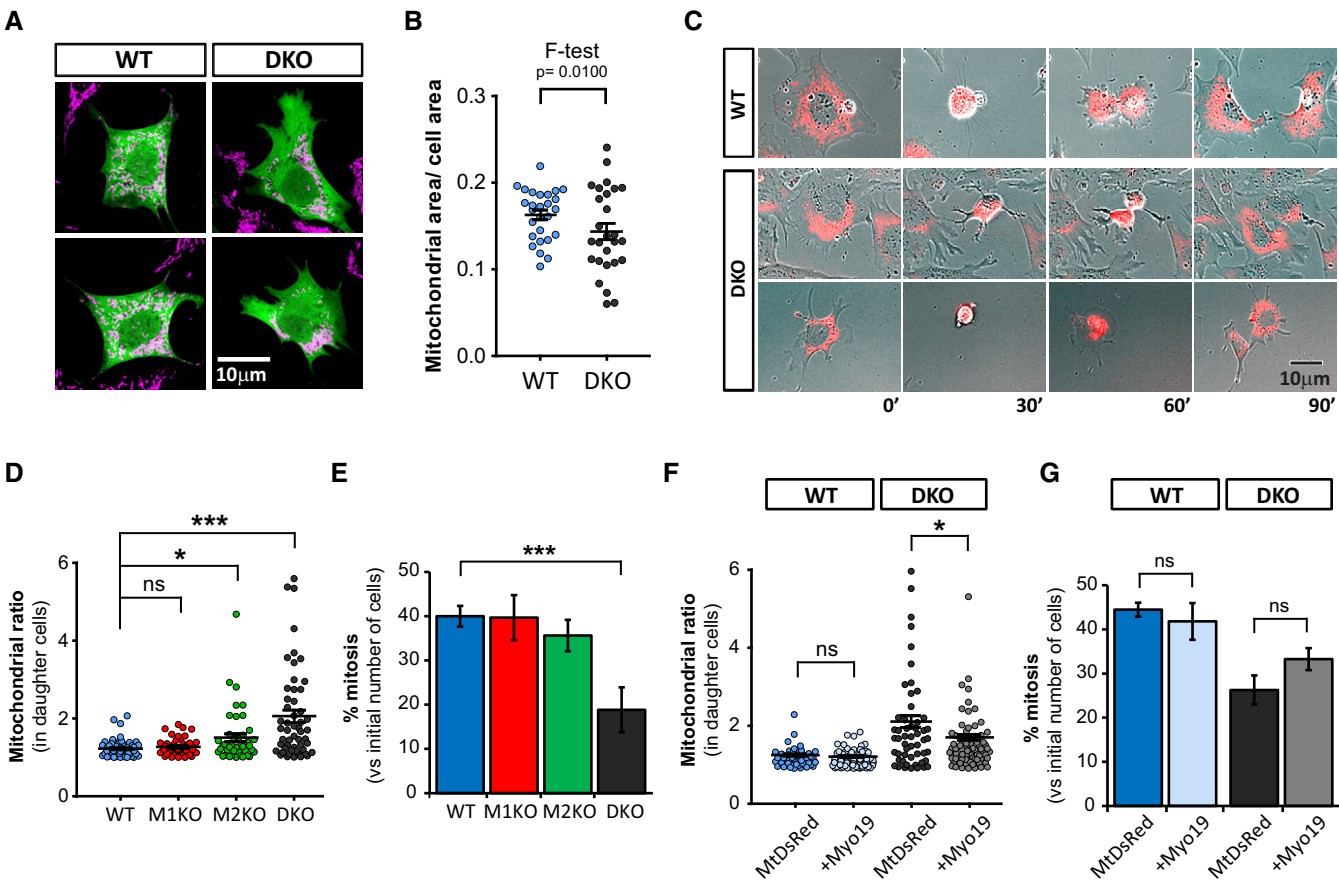

**Figure 8. Miro proteins are necessary to regulate mitochondrial segregation during mitosis.**

A  Representative images of two WT and two Miro^DKO cells showing different variability of mitochondrial area per cell. Tom20 (red) was used to reveal mitochondria and GFP (green) to fill the cell.

B  Graph showing the ratio of mitochondria area per cell area (n = number of cells; WT 27, Miro^DKO 27; F-test).

C  Movie stills from a WT and two Miro^DKO cells expressing the mitochondrial marker Su9GFP (in red to enhance contrast with the bright-field image) undergoing mitosis. Often Miro^DKO cell division shows unequal segregation of mitochondria to the daughter cells. Asymmetric distribution of mitochondria is evident in these Miro^DKO cells before mitosis starts.

D  Quantification showing that Miro^DKO cells, and to a lower extent Miro2^KO cells, fail to equally segregate mitochondria between the two daughter cells during mitosis (n = number of mitotic events; WT 67, Miro1^KO 66, Miro2^KO 76, Miro^DKO 53; Kruskal–Wallis–Dunn's correction).

E  Percentage of cells undergoing mitosis during a 10-h period. Values are normalised with the initial number of cells (n = 3 independent experiments; ANOVA-NK).

F  Quantification of mitochondrial segregation in WT and Miro^DKO cells transfected with Su9GFP (control) or MtDsRed + GFPMyo19 showing that Myo19 partially rescues equal mitochondrial segregation during mitosis (n = number of mitotic events; WT (Su9GFP) 55, WT (GFPMyo19) 77, Miro^DKO (Su9GFP) 54, Miro^DKO (GFPMyo19) 76; Kruskal–Wallis test with Dunn's correction).

G  Percentage of cells undergoing mitosis from videos used in (F). Values are normalised with the initial number of cells (n = 3 independent experiments; ANOVA-NK).

Data information: Error bars represent s.e.m. Statistical significance: *P < 0.05 and ***P < 0.001.

embryos suggests that one copy of Miro1 is sufficient to allow development to continue for a further 4 days (compared to the Miro^DKO) but cannot compensate for all Miro functions at this stage. Remarkably, lethality in the Miro1^het/Miro2^KO animals occurs at a similar stage to Drp1 knockout embryos, which die between E10.5 and E12.5 due to developmental abnormalities including altered forebrain development (Ishihara et al, 2009; Wakabayashi et al, 2009). Miro proteins are, therefore, critical for the normal patterning and growth during early embryonic development, probably via their core conserved functions in regulating mitochondrial positioning and downstream functions such as cell division in addition to later roles in neuronal development and plasticity (Lopez-Domenech et al, 2016; Vaccaro et al, 2017).

Generating MEF lines from the different genotypes of Miro KO embryos allowed us to explore the involvement of Miro proteins in actin- and microtubule-dependent mitochondrial trafficking and positioning. Using substrate micropatterning, we generated a quantitative "reference" distribution map of positioning of the mitochondrial network upon knockout of Miro1, Miro2 or both Miro proteins. Although knocking out Miro1 led to a depletion of mitochondria from the cell periphery, this was greatly enhanced upon removal of both Miro proteins which led to a substantial perinuclear collapse of the mitochondrial network. While knocking out Miro2 had no effect on the Mito^95 value, the additive effect of Miro^DKO suggests that both Miro proteins cooperate to control mitochondrial distribution in the proximal–distal axis within the cell and that Miro1 may be

more effective at compensating for the loss of Miro2 while the reverse is not true. In addition, our live cell imaging experiments also revealed that knockout of either Miro protein could contribute to a significant reduction of mitochondrial displacements within the cell. However, Miro1 appeared to be the primary mediator of long-range mitochondrial trafficking along microtubules, suggesting that Miro2 may be more important for other forms of trafficking. The shorter mitochondria observed in a population of Miro[DKO] cells upon deletion of both Miros is likely a consequence of the dramatically disrupted mitochondrial trafficking, which would likely reduce the number of mitochondrial encounters, and hence the number of fusion events. Our demonstration of a greater impact of Miro1 versus Miro2 knockout for long-range mitochondrial movements agrees with recent work in neurons where Miro1, rather than Miro2, was found to be the primary regulator of long-range mitochondrial transport in axons and dendrites (Lopez-Domenech *et al*, 2016).

A remarkable observation of our experiments is that although long-range transport is considerably disrupted in the absence of all Miro, about 30% of microtubule-dependent runs were preserved in these cells. Our fractionation and imaging experiments show that the kinesin and dynein motors, and the adaptors TRAK1 and TRAK2, can still localise to the mitochondrial membrane in Miro[DKO] cells despite a complete lack of Miro. Moreover, anterograde transport of mitochondria could still be enhanced by TRAK/KIF5 overexpression in Miro[DKO] MEFs, presumably due to the ability of TRAK proteins localised at the OMM to still recruit kinesin motors. However, given the significant disruption of long-range microtubule transport in Miro[DKO] MEFs, TRAK/Kinesin complexes may not be fully functional and may require Miro function to be properly coordinated. These unexpected results challenge the prevailing model that Miro proteins are the obligate acceptor sites on the OMM for the TRAK adaptor proteins (MacAskill & Kittler, 2010; Saxton & Hollenbeck, 2012; Schwarz, 2013; Maeder *et al*, 2014; Mishra & Chan, 2014; Sheng, 2014) and reopens the question about the mechanism by which these cytoplasmic proteins, TRAK1 and TRAK2, localise to the mitochondria to regulate their transport. Our experiments show that at least Mfn1 is one OMM protein with the ability to interact with TRAK1 and TRAK2 on the mitochondria in the complete absence of Miro. This observation is supported by recent work reporting an interaction between TRAK1 and both mitofusins (Lee *et al*, 2017). We cannot rule out that other candidates like Mfn2, DISC1, syntaphilin or the Armcx family of proteins (Misko *et al*, 2010; Lopez-Domenech *et al*, 2012; Chen & Sheng, 2013; Cartoni *et al*, 2016; Norkett *et al*, 2016), all of them shown to regulate mitochondrial transport, may also be able to interact with and recruit TRAK proteins in the absence of Miro.

Our results also support a model whereby TRAK2-mediated retrograde transport is primarily facilitated by Miro1. TRAK2 preferentially binds to dynein/dynactin to mediate the targeting of mitochondria to the dendritic compartment in neurons (van Spronsen *et al*, 2013). In our experiments, TRAK2 can favour retrograde transport (and oppose kinesin-mediated anterograde transport) when Miro1 is present. Indeed, in the absence of Miro1, the TRAK2 effect switches to favour anterograde trafficking, suggesting a regulatory role for Miro1 in the coordination of the balance between dynein and kinesin activities.

Mitochondria can also be trafficked along actin filaments through the unconventional myosin motor Myo19 (Quintero *et al*, 2009),

which may be important in mediating short-range movements and for anchoring mitochondria to the actin cytoskeleton. Interestingly, we found that Miro proteins interact with Myo19 and that Miro depletion leads to an almost complete loss of Myo19 from mitochondria in Miro[DKO] cells. Myo19 can localise to the mitochondria through its positively charged MyMOMA domain (Hawthorne *et al*, 2016) and we show that this mitochondrial targeting is not affected by the lack of Miro as exogenous GFPMyo19 can still be localised in Miro[DKO] mitochondria. However, our data demonstrate that Miro proteins are critical for regulating Myo19 recruitment and stability on the OMM. The almost complete loss of Myo19 in the absence of Miro suggests that Myo19 is highly labile and a target for rapid proteosomal degradation. We also noted that the single knockout for Miro2 had a stronger effect than knockout for Miro1 on endogenous Myo19 mitochondrial stability as shown by the reduced levels of Myo19 in Miro2[KO] cells, suggesting that Miro2 is more efficient in protecting Myo19 from degradation. This may explain why Miro2[KO] cells show a reduction in mitochondrial displacements and contribute to a compound effect of Miro[DKO] on mitochondrial trafficking and distribution (presumably through Myo19-mediated movements) even though Miro1 is the primary mediator of longer range microtubule-dependent transport.

Our findings also provide new insights regarding altered mitochondrial trafficking and cytoskeletal anchoring during mitochondrial quality control. Upon mitochondrial damage, Miro proteins are rapidly ubiquitinated by the PINK1/Parkin mitophagy pathway (Wang *et al*, 2011; Birsa *et al*, 2014) to arrest mitochondrial transport. Since Miro proteins are critical for stabilising Myo19 levels on the mitochondria, Miro loss upon mitochondrial damage leads to a rapid loss of Myo19 from the outer mitochondrial membrane. These results suggest that an important consequence of PINK1/Parkin-mediated Miro degradation is the uncoupling of mitochondria from an actin-dependent trafficking and/or anchorage via Myo19 degradation. Thus, uncoupling mitochondria from the actin cytoskeleton via myosin degradation may be an important step in early stages of the mitophagic process.

Interestingly, we also observed that the collapse of mitochondria upon complete Miro loss of function greatly impairs the balanced segregation of the mitochondrial population to daughter cells during cell division, which correlates with a significantly reduced cell division rate and with a slightly increased death rate in Miro[DKO] MEFs. As we demonstrate in this study, Miro[DKO] MEFs are also loss of function for mitochondrial Myo19. Our results are in agreement with the recent demonstration that Myo19 is important for correct mitochondrial segregation during cell division (Rohn *et al*, 2014). Given the dynamic nature of Myo19 stability on the mitochondrial membrane, it will be interesting in the future to determine whether Myo19 expression levels can be rapidly regulated during the cell cycle and whether this may also be due to Miro regulation. Indeed, Miro proteins may play a central coordinating role in this process through fine-tuning the balance of actin- and microtubule-dependent mitochondrial positioning for correct mitochondrial segregation, a process known to be dependent on Myo19 and in the shedding of microtubule motors from the mitochondria (Rohn *et al*, 2014; Chung *et al*, 2016). The fact that exogenous Myo19 can partially rescue mitochondrial segregation supports this view and indicates that Miro proteins may be necessary to coordinate both Myo19 stabilisation and activity with microtubule-dependent mitochondrial positioning

or motility. This central coordinating role is supported by our studies of embryonic development. Early stages of development are characterised by a high proliferative rate. The altered mitochondrial segregation to daughter cells and associated slower mitosis rate may in part contribute to the observed embryonic lethality in Miro1[het]/Miro2[KO] at E12.5 and in Miro[DKO] embryos at E10.5.

Our work suggests Miro proteins play a central role in coordinating microtubule- and actin-dependent forces on positioning of mitochondria in cells. By coupling to Myo19 in addition to kinesin and dynein, Miro proteins regulate mitochondrial distribution within the cell using both the actin and microtubule cytoskeleton. Whereas both Miro proteins can act to coordinate microtubule-dependent and actin-dependent mitochondrial trafficking, Miro1 preferentially acts to control microtubule-dependent trafficking through kinesin and dynein, while Miro2 plays a more prominent role in coordinating mitochondrial interactions with the actin cytoskeleton through a more efficient recruitment and stability of endogenous Myo19 in the mitochondrial membrane. This fine balance of mitochondrial positioning, mediated by the concerted action of Miro proteins, plays a central role in embryonic development and key cellular processes such as cell division.

# Materials and Methods

### Generation of mouse embryonic fibroblasts (MEFs)

E8.5 embryos were harvested on ice-cold dissection buffer. Yolk sacs were collected and used for genotyping. The heads and viscera were removed from the embryos, and the remaining tissue was gently triturated by pipetting repeatedly (5–10 strokes) in DMEM complete medium. Cells were then plated on 24-well plates. After several days in culture, cells were immortalised by transfection of the simian virus 40 (SV40) T antigen. After 5–7 passages at low density, the cultures presented a homogeneous cell population and started to grow steadily. Transformed cell lines were then genotyped to confirm genetic background. All experiments were carried out after at least eight passages from the immortalisation process.

### Mitochondrial fractionation

We followed a previously published protocol with minor modifications (Frezza *et al*, 2007). Briefly, MEF cells were grown in 15-cm-diameter culture dishes until 95% confluent. Cells were obtained by trypsinisation, washed twice in sterile PBS and resuspended in isolation buffer (IBc: 10 mM Tris/MOPS; 1 mM EGTA/Tris; 0.2 M glucose; pH 7.4). Cell suspension was homogenised using a Teflon Potter Elvehjem and precipitated twice by centrifugation at 600 *g* for 5 min to remove unbroken cells. Supernatant was subjected to another centrifugation at 7,000 *g* for 10 min to obtain the mitochondrial fraction. Supernatant was further cleared and kept as cytoplasmic fraction.

### Experiments on micropatterns

Mouse embryonic fibroblasts cells from the different genotypes either untransfected or overexpressing the indicated constructs were seeded onto adhesive micropatterned coverslips (CYTOO) at 15,000–20,000 cells/cm². Cells were allowed to attach to the

permissive substrate for 4 h and then fixed with PFA 4%. Fixed coverslips were processed for immunocytochemistry immediately after and appropriately stored before imaging.

### Image acquisition and analysis

*Mitochondrial distribution analysis*
Cells growing on "Y"-shaped micropatterned substrates were selected on the basis of shape (visualised with phalloidin or a reporter fill) and nuclei (visualised with DAPI) to ensure that only single cells properly attached to the micropatterns were used. The mitochondrial channel (visualised with MitoTracker) was not used to select cells before the acquisition. The mitochondrial area was thresholded, and Sholl analysis of mitochondrial distribution was performed using a custom-made ImageJ plugin (Lopez-Domenech *et al*, 2016). Mitochondrial signal was quantified within shells radiating out from the soma at 1-μm intervals and the cumulative distribution of mitochondrial signal or Mitochondrial Probability Map (MPM) was plotted per genotype. The distance at which 50, 90 and 95% of the total mitochondrial mass is found (Mito[50], Mito[90] and Mito[95] values, respectively) was calculated per each cell by interpolation. One average, Mito[95] value was calculated per genotype per experiment and used to quantify a final Mito[95] value (*n* = experiments) in figures. In case of overexpression experiments, total number of cells was used as the *n* (*n* = cells).

### Statistical analysis

Excel Software (Microsoft) and GraphPad Prism (GraphPad Software, Inc) were used to analyse the data. D'Agostino and Pearson omnibus test was applied when appropriate to test for normality. Student's *t*-test or Mann–Whitney test was used to test differences between two conditions. Statistical differences between multiple conditions of non-parametric data were calculated using Kruskal–Wallis test followed by *post hoc* Dunn's correction. Comparison of multiple conditions with normally distributed data was performed by one-way ANOVA followed by *post hoc* Newman–Keuls test. To test equality of variances, we used the *F*-test for two conditions or Barlett's test for more than two conditions. Statistical significance was fixed at $P < 0.05$, represented as $*P < 0.05$; $**P < 0.01$ and $***P < 0.001$. All values in text are given as average $\pm$ s.e.m. Error bars are s.e.m.

Further experimental details regarding reagents, animals, cell culture and transfection, immunoblotting and immunofluorescence, immunoprecipitation, proximity ligation assays, respirometry and image acquisition and analysis can be found in the Appendix Supplementary Methods section accompanying this manuscript.

Expanded View for this article is available online.

### Acknowledgements

We thank Prof. Michael Duchen and Dr. Vassilios Kotiadis for their help and assistance with the respirometry experiments. We thank all members of the Kittler laboratory for helpful discussions and comments on the manuscript. This work has been funded by an ERC starting grant (282430), research prize from Lister Institute for Preventive Medicine and grants from Medical Research Council to J.T.K. C.C.-C. is in the MRC LMCB PhD programme at UCL (MRC studentship 1368635). R.N. was an MRC CASE Award Ph.D. student. D.F.S. was

    

on the UCL CoMPLEX PhD programme. D.I. was funded by a Brain Research Trust PhD Scholarship on the UCL Clinical Neuroscience Program. E.F.H. was the recipient of a Marie Skłodowska-Curie grant (661733).

## Author contributions

GL-D and JTK conceived and designed the project. GL-D, CC-C, DI, EFH and JTK designed experiments. GL-D, CC-C, DI, EFH, RN and NB performed experiments and analysed the data. DFS developed analytical tools. GL-D and JTK wrote the manuscript. JTK supervised the project. All authors critically read and approved the submitted manuscript.

## Conflict of interest

The authors declare that they have no conflict of interest.

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
