## [Review Process File · The EMBO Journal]

Miro proteins coordinate microtubule- and actin-dependent mitochondrial transport and distribution

Guillermo López-Doménech, Christian Covill-Cooke, Davor Ivankovic, Els F. Halff, David F. Sheehan, Rosalind Norkett, Nicol Birsa and Josef T. Kittler

Review timeline:

Submission date:	22 December 2016
Preliminary Editorial Decision:	02 February 2017
Preliminary Revision	21 February 2017
1st Editorial Decision:	22 February 2017
Revision received:	30 October 2017
Editorial Decision:	15 November 2017
Revision received:	24 November 2017
Accepted:	04 December 2017

Editor:

Transaction Report:

Preliminary Editorial Decision

02 February 2017

Thank you for submitting your manuscript to The EMBO Journal. It has now been seen by three referees whose comments are included below.

As you will see from the reports, all referees express interest in the work and the presented topic. However, all referees also express the opinion that currently the manuscript is rather preliminary, and that quite extensive additional work is needed to provide convincing support to the proposed mechanism.

Therefore, before taking a decision, I think it would be most productive if you could provide me upfront with a point-by-point response to the raised criticisms in order to see how you would address them. Please keep in mind that although our standard revision period is three months, we can extend it to six months in case of more extensive revision.

Please feel free to contact me if you have any questions regarding this pre-decision consultation approach. I'm looking forward to your response.

Referee #1:

In this manuscript the authors present studies addressing the roles of Miro 1 and 2 in regulating mitochondrial morphology and dynamics. The authors use a KO strategy to analyze the roles of Miro 1 and 2 in embryonic development (lethality more accurately), and mitochondria function, morphology and distribution in cultured KO fibroblasts. The main observations are quite interesting. Counter to the current dogma double KO of both Miro proteins does not block the targeting of TRAKs to the mitochondria, indicating the existence of additional mitochondrial binding partners for TRAKs.

The authors also present evidence that Miro binds Myosin 19, recruiting it to the mitochondria and also suppressing its degradation thereby controlling total Myo19 levels in the cell. Concerns with the current form of the manuscript are presented below for the author's consideration.

The title to the first section of the results is not considered an adequate description of what is presented. The observations do not reveal different "roles" but different requirements for embryonic survival of the two Miro. While somewhat semantic, I think most readers would perceive "role" to mean function and the observations to not show specific functions but address the timing of lethality.

Fig 3C presents images showing some alignment of mitochondria with microtubules in MiroDKO cells. Fig 3D shows cells treated with vinblastine to depolymerize microtubules. The authors state that the latter reveals a disruption of the alignment with microtubules in this condition. This statement seems to reflect an obvious necessity as microtubules have been depolymerized and therefore the alignment of mitochondria with microtubules can't exist. The statement of a loss of alignment with microtubules could only be made if there were microtubules present and mitochondria were noted to not align with the microtubules, which clearly can't be the case. Panel 3E on the other hand provides compelling evidence for microtubule based runs by directly addressing the issue with and without vinblastine treatment. In light of this experiment the value of panel 3D then arises as demonstrating the absence of microtubules and uniform tubulin staining in the cytoplasm as a positive control for the effects of vinblastine on microtubules, which is otherwise absent. It is suggested the authors reconsider/remove the initial interpretation of the vinblastine experiment as addressing the alignment of mitochondria with microtubules.

Fig 4A. It appears, in the blots shown, that in the Miro KOs there may be more TRAK1 in the mitochondrial fraction than in the WTs (eyeballing based on the levels of GAPDH as reference baselines for protein loading). e.g., The Miro1 KO has less GAPDH than WT but a much darker band for TRAK1. I would suggest the authors consider this issue and provide a quantitative analysis.

Fig 4. Untransfected cells are strictly speaking not the appropriate control for the experiment transfecting TRAK1. Since a subtle redistribution of mitochondria is concluded from comparison of TRAK1 transfection to untransfected, the authors need to control for the effect of transfection alone vs the non-transfected control. A simple comparison of the relevant metrics in non-transfected to control transfected cells should remove all concerns regarding the absence of this control, if indeed there is no effect of transfection alone.

Fig 6A and C show an almost complete absence of myo19 by western in all cellular fractions analyzed by western and also by immunocytochemistry, respectively. This is very different from the interpretation initially presented in the text that in the absence of one or the other Miro there is less Myo19 at mitochondria, and almost none in the double KO. Rather, there seems to be a loss of myo19 in and throughout cells.

The authors arrive at this conclusion regarding the regulation of Myo19 levels in the next panels of Fig 6. However, as presented the reader gets the impression that the initial aspects of Fig 6 show specific effects on mitochondrial localization of myo19, which is not the case as the effect is on total myo19 levels and consequently resulting in decreased myo19 being present at mitochondria (and indeed elsewhere). Thus, it is suggested that the presentation of the data in Fig 6 be revised to first note that myo19 levels are impacted by Miro KD. Then, presentation of the experimental evidence for the regulation of myo19 levels, and finally that, as then predicted by the strong decrease in overall myo19 levels, there is correspondingly, if not expectedly, less myo19 at mitochondria. As presented, it seems a case of putting the cart before the horse.

pg 20. The authors write that Miro plays a central role in coordinating "opposing" actions of microtubule and actin-dependent forces. It is not clear how the data show "opposing" forces/functions for microtubule vs actin based mechanisms. The second sentence in this paragraph seems a better description of the observations.

Referee #2:

By generating Miro1/2 KO embryos and MEFs, López-Doménech et al (EMBOJ-2016-96380)

report an unexpected role of mammalian Miro1/2 proteins in coordinating mitochondrial distribution through both microtubule- and actin-based motor mechanisms. This is a follow-up study from their recent work (López-Doménech et al., 2016) in cKO mouse models that show Miro1 as the primary regulator of mitochondrial transport in neuronal axons and dendrites. This study is also supported by an early mouse genetic study demonstrating that a neuron-specific loss of Miro1 causes the depletion of mitochondria from corticospinal tract axons and results in progressive neurological deficits (Nguyen et al., 2014). These studies call for an urgent investigation into whether mammalian Miro1 and Miro2 have overlapping or complementary functions and whether the total loss of Miro1/2 would still permit any mitochondrial transport in mammalian cells.

Using a substrate micro-patterning culture system, the authors generated a quantitative mitochondrial distribution map upon knockout of Miro1, Miro2 or both Miro1/2 in MEFs. By live imaging, they revealed that Miro1 appeared to be the primary mediator of mitochondrial trafficking along MTs. Surprisingly, they found that about 30% of MT-based mitochondrial movements were preserved in Miro1/2 DKO cells. The fractionation and imaging experiments further demonstrate that both kinesin and dynein motors, along with their motor adaptors TRAK1 and TRAK2, can still be associated with mitochondria despite a complete loss of Miro1/2. It is even more surprising that mitochondrial anterograde transport is enhanced by overexpression of TRAK and KIF5 in DKO MEFs. In addition, Miro proteins recruit Myo19 onto the mitochondrial outer membrane; these roles are critical for symmetric mitochondrial segregation during cell division. Based on these findings, the authors propose a new attractive model: Miro proteins play a central role in coordinating MT- and actin-dependent trafficking and positioning of mitochondria. These unexpected results challenge the current model that Miro proteins are the essential receptors for selectively recruiting kinesin-TRAK motor-adaptor complexes to mitochondria.

Overall, this is an interesting study, which could potentially make an important contribution to the mitochondrial transport field. The reported results are highly unexpected, thus, the study will fill a gap in our current knowledge and is suitable for the broad readership of the EMBO J. The manuscript is logically presented and most of the data are solid and in general of excellent quality. It represents a great deal of work, as evidenced by the comprehensive single and double KO analysis in mouse embryos and MEFs.

Main concerns:

The current study, however, is rather descriptive. It lacks cellular mechanistic insights into the observed phenotypes. For example: (1) How are TRAK1/2 recruited to mitochondria in the absence of Miro1/2? (2) How does Miro1 selectively facilitate TRAK2-dependent retrograde transport of mitochondria? (3) Is Miro-mediated recruitment / stability of Myo19 onto mitochondria sufficient to drive actin-based movement or actin-mediated anchoring of mitochondria? If this is the case, (4) how could these coupling defects impact mitochondrial segregation. Addressing one or two of these mechanistic questions would make this study more compelling not only to challenge the current models, but also to advance our knowledge of the Miro-TRAK-motor complexes for their coordination of mitochondrial trafficking and anchoring by engaging on MT and actin.

If the Miro proteins act as acceptors for Myo19, one should see the remaining Myo19 in the cytosol of DKO cells. It is difficult to interpret why almost all of Myo19 is lost in cell lysates and cytoplasmic fractions of DKO cells (Fig 6A and 6B). In addition, I am curious to know how the authors obtained the data showing the relative enrichment of Myo19 on mitochondria by calculating as mitochondrial signal / cytoplasmic signal. Again, the same concern applies to the imaging data (Fig 6C, 6D), where the Myo19 signal is lost globally in DKO cells, but not necessary to mitochondria-labeled Myo19. Based on these data, one would propose an alternative role of Miro in either regulating Myo19 expression or maintaining its stability. This new role of Miro is quite interesting considering its impact on actin-based movement versus anchoring, and thus presumably regulating mitochondrial segregation. However, the current study makes no effort to explore the possible mechanisms.

Other minor points:

Page 8: The statement: "The prevailing model of Miro function is that it acts as the essential adaptor for recruiting the motor complexes to the mitochondria to drive mitochondrial transport along the microtubule tracks". It would be more accurate to state: "...as the essential receptor for recruiting

the motor-adaptor complexes.....".

Video 3: The majority of mitochondria in DKO are still in a tubular shape, which is not consistent with what is shown in Fig. 2A, 2B.

Fig 3 and Videos 1, 3: I am surprised by the quantitative data that shows such a small number of motile mitochondria associated with MT-based transport in each cell. In the video, I could easily see more mitochondria in WT cells undergoing such movement.

Fig 4A: Fractions "I, C, M" should be noted in the fig legend. In Fig 4B, higher magnification images would be helpful to see the mitochondrial targeting of TRAKs.

Referee #3:

The manuscript by Lopez-Domenech et al. investigates the role of Miro proteins in mitochondrial trafficking in MEFs generated from Miro1-, Miro2-, and double-knockout mice. The authors examine mitochondrial distribution in these MEFs, and the association of Trak1 and Trak2 as well as motors with mitochondria in these cells. Finally, the authors provide evidence that Miro1/2 double knockout interferes with the proper segregation of mitochondria to daughter cells during cytokinesis

The generation of Miro knockout mice, first described in the paper from the same group last year (Lopez-Domenech et al., 2016) provides important models that are further investigated here. Previously, the authors studied the effects of Miro1 and Miro2 knockout on mitochondrial motility in neurons, here they focus on MEFs, and also on cells from double knockout animals.

The most interesting and novel observation from these follow-up studies is that both TRAK proteins and microtubule motors are apparently still recruited to mitochondria in the absence of Miro proteins, suggesting that the canonical role for Miro in TRAK/motor recruitment is unlikely to be true. However, the first sentence of the abstract states that "Miro1 and Miro2 . . . regulate mitochondrial trafficking along microtubules by linking mitochondria to kinesin and dynein motors". This statement suggests that the authors do not yet believe their own data. Part of the problem may be that the authors examine a number of different aspects of mitochondrial trafficking, but do not go far enough in depth on any single point to make a clear and compelling case. Instead, the manuscript reads like two or three separate stories put together into a single manuscript. Overall the work feels fragmented with no single point explored in sufficient detail to make a cohesive story.

In addition, the manuscript is sloppy. The phrase "n=cells" is used, presumably as a placeholder for important statistical information that was never filled in. Labels used in figures are not explained in the legends, such as I, C, and M in Figure 4. Figure 1 is cut off at the top. More significantly, the images shown in many panels do not appear to be representative of the overall quantitative conclusions - for example, compare panels C and G in Fig. 4. Additional specifics are given below.

In sum, this work is too preliminary for publication.

Specific comments on Figures:

Figure 1:

- Panel A-C is cut off on the top.
- It would be nice to have examples of M1het/M2KO, M2KO, and WT at all three developmental stages for comparison
- Panel E the scale bar is mislabeled (presumably that is not 500nm) and the arrows are not described in the figure legend
- In Panel F the order of mutants is confusing and is not consistent between blots F and G. This certainly does not invalidate the findings, but it makes it very difficult to follow.

Figure 2:

- Throughout the manuscript the authors describe using two different cell lines per genotype. While three independent experiments are carried out on these MEFs, this is nonetheless only two biological replicates.

- The authors should consider using a color scheme more friendly to colorblind readers.
- Figure C does not have a scale bar in the detail column

Figure 3:

- The technique used to describe mitochondrial displacement is not clearly described in the legend
- In panel A, the double knockout cell does not appear to have the perinuclear clustering phenotype described in figure 2
- In panel C the mitochondria in the double knockout cell do not appear to display the fragmentation described in figure 2
- In panel D Vinblastine is incorrectly spelled as "vinblastin" and the concentration reads " μm " instead of μM .
- Again, in panel F the mitochondria in the DKO column appear more elongated than the mitochondria in the WT column (in contrast to the claims from fig 2.).

Figure 4:

- This figure demonstrates a new and potential exciting observation. Namely, Trak1, Trak2, and KHC are still recruited to mitochondria in the absence of Miro1 and 2. However, no quantitative analysis is provided.
- Importantly, these proteins not only seem to be recruited to the mitochondria, but their localization to mitochondria appears to be unregulated in the mutants.
- However, the authors fail to address the significance of the apparent upregulation of Trak1 and KHC in the mitochondrial fraction of the M1KO M2KO and DKO cells in panel A.
- This is particularly important, as the subsequent experiments in figure 4 and 5 rely on overexpression of exogenous forms of these proteins on top of already altered protein levels in the mitochondria of the various mutants.
- Quantitation of the western blot from at least three independent biological replicates would be informative, in order to clarify TRAK1/2 and KIF5c levels on mitochondria.
- The authors should consider depleting endogenous TRAK1/2 in the mutant cell lines
- In panels C-G, it appears as though the representative images in C do not convincingly reflect the quantitative data in G. For example, the TRAK2 column in panel C is almost completely absent of mitochondria, while the TRAK2 + KIF5C representative image shows clear mitochondria in the enlarged inset. This appears to be at odds with panel G, which indicates no significant difference between these two conditions. This casts doubt on several of the effects reported.

Figure 6:

- Again, this is interesting that myosin 19 fails to bind to mitochondria in the absence of Miro1/2. Further, panel E nicely demonstrates that rescue with either Miro can recover Myo19 localization to mitochondria
- It would be nice if the authors discussed the Miro dependent mitochondrial localization of Myo19 in light of the findings from Hawthorne et al. (2016, Cytoskeleton) indicating that Myo19 is targeted to the mitochondrial outer membrane by positively charged residues within its MyMOMA domain.
- Mito tracker is spelled wrong in panel E
- Panels H-I appears to be the only evidence substantiating the claim that "Moreover, Miro depletion during PINK1/Parkin-dependent mitophagy can also drive a loss of mitochondrial Myo19 upon mitochondrial damage."
- While the authors show that Myo19 and Miro1 have different degradation kinetics upon FCCP treatment, this does not indicate that Miro depletion is responsible for the loss of Myo19. To show this more clearly, the authors should examine the rate of FCCP driven myo19 degradation in Miro1 KO, and Miro 2 KO cells.
- It would also be nice to see the experiments in H and I performed in the MEFs described throughout the paper
- Additionally, the authors should examine the degradation kinetics of other outer membrane markers such as TOMM20 or Mitofusin 1.
- Finally, the experiment should be repeated in the presence of a proteasome inhibitor such as MG132 to determine whether Myo19 and Miro degradation is driven by the proteasome or autophagic degradation.

Figure 7:

- The authors show that mitochondria are unequally segregated between daughter cells in the Miro1/2 DKO. This observation suggests that at least one Miro is required for appropriate

mitochondrial network distribution during cytokinesis.

- What are the functional consequences of this asymmetrical distribution of mitochondria?
- Do you observe recovery of mitochondrial mass in the cells that receive a smaller proportion of mitochondria from the mother cell?
- Do you observe an increase in multinucleate cells in the DKO cells similar to what was described by Rohn et al (2014 Current biology) in Myo19 KD cells.
- Throughout the paper sample sizes are not properly indicated (often n values just say n=cells)
- For the SHOLL analysis, the authors do not report the radii of each concentric circle, nor do they describe the average radii of the nuclei of each cell.

Preliminary Revision - authors' response

21 February 2017

Referee#1 has only relatively minor technical and presentational concerns that we think will be straightforward to address and also requests further quantification of some fractionation experiments (also requested by Referee#3), which we can readily provide. In addition to a number of technical issues Referee#2 also requests us to provide 'one or two' pieces of additional mechanistic insight including providing more insight into the mechanisms by which TRAKs can localize to the mitochondria in the absence of Miro and how Miro proteins regulate Myo19. Finally Referee#3 has a number of technical concerns which we will address and also raises similar questions to Referee#2 relating to providing more insight into the regulatory cross-talk between Miro proteins and Myo19.

We are aware that the three referees have all requested a more in depth characterization of the relationship between Miro and Myo19 and the functional consequences for the regulation of an actin mediated movement/anchoring of mitochondria. As detailed below in a point-by-point response we will pursue a more mechanistic insight of the interaction between Myo19 and Miro. This would include investigating the impact of overexpressing Myo19 or artificially targeting it to the mitochondria in our MEF cell lines in addition to looking at whether re-distributing Miro to another cellular compartment or overexpressing cytoplasmic (lacking its TM) Miro will also re-localise Myo19 and/or alter its stability. We will also further characterize the stability and turnover of Myo19 in the cytoplasm and mitochondrial membrane in the absence of Miro proteins (see point-by-point response for details). We will additionally attempt to test whether Myo19 targeted to mitochondria is readily functional or whether Miro proteins are needed to regulate Myo19 function. We will use our analysis of mitochondrial distribution in addition to segregation during mitosis as read outs of such a functionality.

In addition we will also attempt to provide more insight into how TRAK proteins are still recruited to the mitochondrial membrane in the absence of Miro with the aim of providing more mechanistic information about the nature of the TRAK1 and TRAK2 acceptors in the mitochondrial membrane as suggested by Referee#2. We will determine if candidate mitochondrial TRAK binding proteins are still on the mitochondria in the absence of Miro and if TRAKs can still interact with them. These experiments may also shed light on potential mechanisms of Miro1 specific TRAK2-dependent retrograde transport since the role of Miro proteins may be to regulate the function of other TRAK complexes on the mitochondria (e.g. by positively or negatively enhancing interactions). Thus we feel that characterising other key TRAK complexes on the mitochondria with or without Miro present may provide further mechanistic insights.

Finally we will take advantage of our long term imaging assay to further investigate the mechanisms and impact of disrupted symmetrical mitochondrial segregation during mitosis. In part these experiments may be informed by the programme of work described above and in the point-by-point response. For example the impact of artificially targeting Myo19 to the mitochondria in Miro DKO cells will be determined on mitochondrial segregation. In addition to address Referee#3's query regarding the physiological impact of altered mitochondrial segregation we propose to perform longer term imaging experiments to allow us to directly follow cells that have received a reduced (or augmented) mitochondrial load and determine the impact on growth, cell division and cell death rates.

Thank you for submitting your manuscript for consideration by the EMBO Journal. We have now received three referee reports on your manuscript, which are included below for your information.

Based on the referees' comments and the revision outline you provided during the pre-decision discussion, I would like to invite you to submit a revised version of the manuscript, addressing the comments of all three reviewers. Particularly I would like to ask you to focus on the following points:

- Further characterisation of Myo19 stabilisation and mitochondrial recruitment by Miro (all referees)
- Analysis of TRAK1/2 recruitment to mitochondria in the absence of Miro (all referees)
- Provide a more detailed description of data quantification and statistics, as requested by referees #2 and #3

I should add that it is The EMBO Journal policy to allow only a single major round of revision and that it is therefore important to resolve the main concerns at this stage.

We generally allow three months as standard revision time, but an extension to six months is possible in case of an extensive revision.

Please feel free to contact me with any further questions regarding the revision. Thank you for the opportunity to consider your work for publication. I look forward to your revision.

Referee #1:

In this manuscript the authors present studies addressing the roles of Miro 1 and 2 in regulating mitochondrial morphology and dynamics. The authors use a KO strategy to analyze the roles of Miro 1 and 2 in embryonic development (lethality more accurately), and mitochondria function, morphology and distribution in cultured KO fibroblasts. The main observations are quite interesting. Counter to the current dogma double KO of both Miro's does not block the targeting of TRAKs to the mitochondria, indicating the existence of additional mitochondrial binding partners for TRAKs. The authors also present evidence that Miro binds Myosin 19, recruiting it to the mitochondria and also suppressing its degradation thereby controlling total Myo19 levels in the cell. Concerns with the current form of the manuscript are presented below for the author's consideration.

The title to the first section of the results is not considered an adequate description of what is presented. The observations do not reveal different "roles" but different requirements for embryonic survival of the two Miro. While somewhat semantic, I think most readers would perceive "role" to mean function and the observations to not show specific functions but address the timing of lethality.

Fig 3C presents images showing some alignment of mitochondria with microtubules in MiroDKO cells. Fig 3D shows cells treated with vinblastine to depolymerize microtubules. The authors state that the latter reveals a disruption of the alignment with microtubules in this condition. This statement seems to reflect an obvious necessity as microtubules have been depolymerized and therefore the alignment of mitochondria with microtubules can't exist. The statement of a loss of alignment with microtubules could only be made if there were microtubules present and mitochondria were noted to not align with the microtubules, which clearly can't be the case. Panel 3E on the other hand provides compelling evidence for microtubule based runs by directly addressing the issue with and without vinblastine treatment. In light of this experiment the value of panel 3D then arises as demonstrating the absence of microtubules and uniform tubulin staining in the cytoplasm as a positive control for the effects of vinblastine on microtubules, which is otherwise absent. It is suggested the authors reconsider/remove the initial interpretation of the vinblastine experiment as addressing the alignment of mitochondria with microtubules.

Fig 4A. It appears, in the blots shown, that in the Miro KOs there may be more TRAK1 in the mitochondrial fraction than in the WTs (eyeballing based on the levels of GAPDH as reference baselines for protein loading). e.g., The Miro1 KO has less GAPDH than WT but a much darker band for TRAK1. I would suggest the authors consider this issue and provide a quantitative analysis.

Fig 4. Untransfected cells are strictly speaking not the appropriate control for the experiment transfecting TRAK1. Since a subtle redistribution of mitochondria is concluded from comparison of TRAK1 transfection to untransfected, the authors need to control for the effect of transfection alone vs the non-transfected control. A simple comparison of the relevant metrics in non-transfected to control transfected cells should remove all concerns regarding the absence of this control, if indeed there is no effect of transfection alone.

Fig 6A and C show an almost complete absence of myo19 by western in all cellular fractions analyzed by western and also by immunocytochemistry, respectively. This is very different from the interpretation initially presented in the text that in the absence of one or the other Miro there is less Myo19 at mitochondria, and almost none in the double KO. Rather, there seems to be a loss of myo19 in and throughout cells.

The authors arrive at this conclusion regarding the regulation of Myo19 levels in the next panels of Fig 6. However, as presented the reader gets the impression that the initial aspects of Fig 6 show specific effects on mitochondrial localization of myo19, which is not the case as the effect is on total myo19 levels and consequently resulting in decreased myo19 being present at mitochondria (and indeed elsewhere). Thus, it is suggested that the presentation of the data in Fig 6 be revised to first note that myo19 levels are impacted by Miro KD. Then, presentation of the experimental evidence for the regulation of myo19 levels, and finally that, as then predicted by the strong decrease in overall myo19 levels, there is correspondingly, if not expectedly, less myo19 at mitochondria. As presented, it seems a case of putting the cart before the horse.

pg 20. The authors write that Miro plays a central role in coordinating "opposing" actions of microtubule and actin-dependent forces. It is not clear how the data show "opposing" forces/functions for microtubule vs actin based mechanisms. The second sentence in this paragraph seems a better description of the observations.

Referee #2:

By generating Miro1/2 KO embryos and MEFs, López-Doménech et al (EMBOJ-2016-96380) report an unexpected role of mammalian Miro1/2 proteins in coordinating mitochondrial distribution through both microtubule- and actin-based motor mechanisms. This is a follow-up study from their recent work (López-Doménech et al., 2016) in cKO mouse models that show Miro1 as the primary regulator of mitochondrial transport in neuronal axons and dendrites. This study is also supported by an early mouse genetic study demonstrating that a neuron-specific loss of Miro1 causes the depletion of mitochondria from corticospinal tract axons and results in progressive neurological deficits (Nguyen et al., 2014). These studies call for an urgent investigation into whether mammalian Miro1 and Miro2 have overlapping or complementary functions and whether the total loss of Miro1/2 would still permit any mitochondrial transport in mammalian cells.

Using a substrate micro-patterning culture system, the authors generated a quantitative mitochondrial distribution map upon knockout of Miro1, Miro2 or both Miro1/2 in MEFs. By live imaging, they revealed that Miro1 appeared to be the primary mediator of mitochondrial trafficking along MTs. Surprisingly, they found that about 30% of MT-based mitochondrial movements were preserved in Miro1/2 DKO cells. The fractionation and imaging experiments further demonstrate that both kinesin and dynein motors, along with their motor adaptors TRAK1 and TRAK2, can still be associated with mitochondria despite a complete loss of Miro1/2. It is even more surprising that mitochondrial anterograde transport is enhanced by overexpression of TRAK and KIF5 in DKO

MEFs. In addition, Miro proteins recruit Myo19 onto the mitochondrial outer membrane; these roles are critical for symmetric mitochondrial segregation during cell division. Based on these findings, the authors propose a new attractive model: Miro proteins play a central role in coordinating MT- and actin-dependent trafficking and positioning of mitochondria. These unexpected results challenge the current model that Miro proteins are the essential receptors for selectively recruiting kinesin-TRAK motor-adaptor complexes to mitochondria.

Overall, this is an interesting study, which could potentially make an important contribution to the mitochondrial transport field. The reported results are highly unexpected, thus, the study will fill a gap in our current knowledge and is suitable for the broad readership of the EMBO J. The manuscript is logically presented and most of the data are solid and in general of excellent quality. It represents a great deal of work, as evidenced by the comprehensive single and double KO analysis in mouse embryos and MEFs.

Main concerns:

The current study, however, is rather descriptive. It lacks cellular mechanistic insights into the observed phenotypes. For example: (1) How are TRAK1/2 recruited to mitochondria in the absence of Miro1/2? (2) How does Miro1 selectively facilitate TRAK2-dependent retrograde transport of mitochondria? (3) Is Miro-mediated recruitment / stability of Myo19 onto mitochondria sufficient to drive actin-based movement or actin-mediated anchoring of mitochondria? If this is the case, (4) how could these coupling defects impact mitochondrial segregation. Addressing one or two of these mechanistic questions would make this study more compelling not only to challenge the current models, but also to advance our knowledge of the Miro-TRAK-motor complexes for their coordination of mitochondrial trafficking and anchoring by engaging on MT and actin.

If the Miro proteins act as acceptors for Myo19, one should see the remaining Myo19 in the cytosol of DKO cells. It is difficult to interpret why almost all of Myo19 is lost in cell lysates and cytoplasmic fractions of DKO cells (Fig 6A and 6B). In addition, I am curious to know how the authors obtained the data showing the relative enrichment of Myo19 on mitochondria by calculating as mitochondrial signal / cytoplasmic signal. Again, the same concern applies to the imaging data (Fig 6C, 6D), where the Myo19 signal is lost globally in DKO cells, but not necessary to mitochondria-labeled Myo19. Based on these data, one would propose an alternative role of Miro in either regulating Myo19 expression or maintaining its stability. This new role of Miro is quite interesting considering its impact on actin-based movement versus anchoring, and thus presumably regulating mitochondrial segregation. However, the current study makes no effort to explore the possible mechanisms.

Other minor points:

Page 8: The statement: "The prevailing model of Miro function is that it acts as the essential adaptor for recruiting the motor complexes to the mitochondria to drive mitochondrial transport along the microtubule tracks". It would be more accurate to state: "...as the essential receptor for recruiting the motor-adaptor complexes.....".

Video 3: The majority of mitochondria in DKO are still in a tubular shape, which is not consistent with what is shown in Fig. 2A, 2B.

Fig 3 and Videos 1, 3: I am surprised by the quantitative data that shows such a small number of motile mitochondria associated with MT-based transport in each cell. In the video, I could easily see more mitochondria in WT cells undergoing such movement.

Fig 4A: Fractions "I, C, M" should be noted in the fig legend. In Fig 4B, higher magnification images would be helpful to see the mitochondrial targeting of TRAKs.

Referee #3:

The manuscript by Lopez-Domenech et al. investigates the role of Miro proteins in mitochondrial trafficking in MEFs generated from Miro1-, Miro2-, and double-knockout mice. The authors

examine mitochondrial distribution in these MEFs, and the association of Trak1 and Trak2 as well as motors with mitochondria in these cells. Finally, the authors provide evidence that Miro1/2 double knockout interferes with the proper segregation of mitochondria to daughter cells during cytokinesis

The generation of Miro knockout mice, first described in the paper from the same group last year (Lopez-Domenech et al., 2016) provides important models that are further investigated here. Previously, the authors studied the effects of Miro1 and Miro2 knockout on mitochondrial motility in neurons, here they focus on MEFs, and also on cells from double knockout animals.

The most interesting and novel observation from these follow-up studies is that both TRAK proteins and microtubule motors are apparently still recruited to mitochondria in the absence of Miro proteins, suggesting that the canonical role for Miro in TRAK/motor recruitment is unlikely to be true. However, the first sentence of the abstract states that "Miro1 and Miro2 . . . regulate mitochondrial trafficking along microtubules by linking mitochondria to kinesin and dynein motors". This statement suggests that the authors do not yet believe their own data. Part of the problem may be that the authors examine a number of different aspects of mitochondrial trafficking, but do not go far enough in depth on any single point to make a clear and compelling case. Instead, the manuscript reads like two or three separate stories put together into a single manuscript. Overall the work feels fragmented with no single point explored in sufficient detail to make a cohesive story.

In addition, the manuscript is sloppy. The phrase "n=cells" is used, presumably as a placeholder for important statistical information that was never filled in. Labels used in figures are not explained in the legends, such as I, C, and M in Figure 4. Figure 1 is cut off at the top. More significantly, the images shown in many panels do not appear to be representative of the overall quantitative conclusions - for example, compare panels C and G in Fig. 4. Additional specifics are given below.

In sum, this work is too preliminary for publication.

Specific comments on Figures:

Figure 1:

- Panel A-C is cut off on the top.
- It would be nice to have examples of M1het/M2KO, M2KO, and WT at all three developmental stages for comparison
- Panel E the scale bar is mislabeled (presumably that is not 500nm) and the arrows are not described in the figure legend
- In Panel F the order of mutants is confusing and is not consistent between blots F and G. This certainly does not invalidate the findings, but it makes it very difficult to follow.

Figure 2:

- Throughout the manuscript the authors describe using two different cell lines per genotype. While three independent experiments are carried out on these MEFs, this is nonetheless only two biological replicates.
- The authors should consider using a color scheme more friendly to colorblind readers.
- Figure C does not have a scale bar in the detail column

Figure 3:

- The technique used to describe mitochondrial displacement is not clearly described in the legend
- In panel A, the double knockout cell does not appear to have the perinuclear clustering phenotype described in figure 2
- In panel C the mitochondria in the double knockout cell do not appear to display the fragmentation described in figure 2
- In panel D Vinblastine is incorrectly spelled as "vinblastin" and the concentration reads " μm " instead of μM .
- Again, in panel F the mitochondria in the DKO column appear more elongated than the mitochondria in the WT column (in contrast to the claims from fig 2.).

Figure 4:

- This figure demonstrates a new and potential exciting observation. Namely, Trak1, Trak2, and KHC are still recruited to mitochondria in the absence of Miro1 and 2. However, no quantitative

analysis is provided.

- Importantly, these proteins not only seem to be recruited to the mitochondria, but their localization to mitochondria appears to be unregulated in the mutants.
- However, the authors fail to address the significance of the apparent upregulation of Trak1 and KHC in the mitochondrial fraction of the M1KO M2KO and DKO cells in panel A.
- This is particularly important, as the subsequent experiments in figure 4 and 5 rely on overexpression of exogenous forms of these proteins on top of already altered protein levels in the mitochondria of the various mutants.
- Quantitation of the western blot from at least three independent biological replicates would be informative, in order to clarify TRAK1/2 and KIF5c levels on mitochondria.
- The authors should consider depleting endogenous TRAK1/2 in the mutant cell lines
- In panels C-G, it appears as though the representative images in C do not convincingly reflect the quantitative data in G. For example, the TRAK2 column in panel C is almost completely absent of mitochondria, while the TRAK2 + KIF5C representative image shows clear mitochondria in the enlarged inset. This appears to be at odds with panel G, which indicates no significant difference between these two conditions. This casts doubt on several of the effects reported.

Figure 6:

- Again, this is interesting that myosin 19 fails to bind to mitochondria in the absence of Miro1/2. Further, panel E nicely demonstrates that rescue with either Miro can recover Myo19 localization to mitochondria
- It would be nice if the authors discussed the Miro dependent mitochondrial localization of Myo19 in light of the findings from Hawthorne et al. (2016, Cytoskeleton) indicating that Myo19 is targeted to the mitochondrial outer membrane by positively charged residues within its MyMOMA domain.
- Mito tracker is spelled wrong in panel E
- Panels H-I appears to be the only evidence substantiating the claim that "Moreover, Miro depletion during PINK1/Parkin-dependent mitophagy can also drive a loss of mitochondrial Myo19 upon mitochondrial damage."
- While the authors show that Myo19 and Miro1 have different degradation kinetics upon FCCP treatment, this does not indicate that Miro depletion is responsible for the loss of Myo19. To show this more clearly, the authors should examine the rate of FCCP driven myo19 degradation in Miro1 KO, and Miro 2 KO cells.
- It would also be nice to see the experiments in H and I performed in the MEFs described throughout the paper
- Additionally, the authors should examine the degradation kinetics of other outer membrane markers such as TOMM20 or Mitofusin 1.
- Finally, the experiment should be repeated in the presence of a proteasome inhibitor such as MG132 to determine whether Myo19 and Miro degradation is driven by the proteasome or autophagic degradation.

Figure 7:

- The authors show that mitochondria are unequally segregated between daughter cells in the Miro1/2 DKO. This observation suggests that at least one Miro is required for appropriate mitochondrial network distribution during cytokinesis.
- What are the functional consequences of this asymmetrical distribution of mitochondria?
- Do you observe recovery of mitochondrial mass in the cells that receive a smaller proportion of mitochondria from the mother cell?
- Do you observe an increase in multinucleate cells in the DKO cells similar to what was described by Rohn et al (2014 Current biology) in Myo19 KD cells.
- Throughout the paper sample sizes are not properly indicated (often n values just say n=cells)
- For the SHOLL analysis, the authors do not report the radii of each concentric circle, nor do they describe the average radii of the nuclei of each cell.

Response to Editors comments

Based on the referees' comments and the revision outline you provided during the pre-decision discussion, I would like to invite you to submit a revised version of the manuscript, addressing the comments of all three reviewers. Particularly I would like to ask you to focus on the following points:

- Further characterisation of Myo19 stabilisation and mitochondrial recruitment by Miro (all referees)
- Analysis of TRAK1/2 recruitment to mitochondria in the absence of Miro (all referees)
- Provide a more detailed description of data quantification and statistics, as requested by referees #2 and #3

We thank the editor for giving us the opportunity to address the reviewers' comments in a revised manuscript. While we have aimed to address all the referees comments as detailed in the point-by-point response below, guided by the editor we have focused in particular on the 3 main points above. In our revised manuscript we now provide a significantly more in depth characterisation of Myo19 recruitment and stabilisation on the mitochondrial membrane by Miro proteins. We now also identify Mitofusin proteins as able to interact with TRAK proteins in the absence of Miro, providing a mechanism whereby they can still be localized to the mitochondrial membrane independent of Miro. Finally, we also provide additional details regarding quantification and statistics.

As part of our characterization of Myo19 stability and its dependency on Miro proteins we now show Myo19 protein levels in our Miro1^{KO}/Miro2^{het} and Miro1^{het}/Miro2^{KO} cell lines (Fig 6A and B). For this reason we have also included these genotypes in the mitochondrial distribution analysis (Fig 2C-E and Fig EV3C, E and F) as a basic characterization of these cell lines.

Response to Referees comments

We are very grateful to all three referees for their helpful comments and suggestions on our manuscript entitled "Miro proteins coordinate microtubule and actin dependent mitochondrial distribution" (EMBOJ-2016-96380), the majority of which we have addressed and which we agree have further improved the manuscript.

Point-by-point response

Referee #1:

In this manuscript the authors present studies addressing the roles of Miro 1 and 2 in regulating mitochondrial morphology and dynamics. The authors use a KO strategy to analyze the roles of Miro 1 and 2 in embryonic development (lethality more accurately), and mitochondria function, morphology and distribution in cultured KO fibroblasts. The main observations are quite interesting. Counter to the current dogma double KO of both Miro proteins does not block the targeting of TRAKs to the mitochondria, indicating the existence of additional mitochondrial binding partners for TRAKs. The authors also present evidence that Miro binds Myosin 19, recruiting it to the mitochondria and also suppressing its degradation thereby controlling total Myo19 levels in the cell. Concerns with the current form of the manuscript are presented below for the author's consideration.

We thank the reviewer for their constructive comments which have significantly helped to improve our manuscript.

1) The title to the first section of the results is not considered an adequate description of what is presented. The observations do not reveal different "roles" but different requirements for embryonic survival of the two Miro. While somewhat semantic, I think most readers would perceive "role" to mean function and the observations to not show specific functions but address the timing of lethality.

We agree with the point that the referee raises. We have changed the title of the first section accordingly. Now it reads as follows: "Differential requirements for Miro1 and Miro2 during embryonic development".

2) Fig 3C presents images showing some alignment of mitochondria with microtubules in MiroDKO cells. Fig 3D shows cells treated with vinblastine to depolymerize microtubules. The authors state that the latter reveals a disruption of the alignment with microtubules in this condition. This statement seems to reflect an obvious necessity as microtubules have been depolymerized and therefore the alignment of mitochondria with microtubules can't exist. The statement of a loss of alignment with microtubules could only be made if there were microtubules present and mitochondria were noted to not align with the microtubules, which clearly can't be the case. Panel 3E on the other hand provides compelling evidence for microtubule based runs by directly addressing the issue with and without vinblastine treatment. In light of this experiment the value of panel 3D then arises as demonstrating the absence of microtubules and uniform tubulin staining in the cytoplasm as a positive control for the effects of vinblastine on microtubules, which is otherwise absent. It is suggested the authors reconsider/remove the initial interpretation of the vinblastine experiment as addressing the alignment of mitochondria with microtubules.

The referee raises a good point and we have followed his/her suggestion. Now both panels have been moved to Appendix Fig S1C and D. As suggested by the referee we now use the panel S1D to provide evidence that microtubules are absent under vinblastine treatment and that our quantification of directional mitochondrial runs account for microtubule dependent movements.

3) Fig 4A. It appears, in the blots shown, that in the Miro KOs there may be more TRAK1 in the mitochondrial fraction than in the WTs (eyeballing based on the levels of GAPDH as reference baselines for protein loading). e.g., The Miro1 KO has less GAPDH than WT but a much darker band for TRAK1. I would suggest the authors consider this issue and provide a quantitative analysis.

This is a good point and was also raised by Referee#3 (Points F4-1 to F4-5). We now provide a quantitative analysis of the motor and adaptor expression levels in the mitochondrial fractions from at least 4 subcellular fractionation experiments. We have included this data in Fig 3, which now contains the mitochondrial motility experiments in MEFs together with the fractionation analysis of the different motor/adaptor proteins in Miro^{DKO} cells. We provide evidence that none of the motor/adaptors tested showed a significant change in the protein levels in mitochondria. We have also taken the opportunity to replace these blots by new ones that better represent our quantitative data and that are loaded in the same order as the Myo19 blot in Fig 6C for consistency.

4) Fig 4. Untransfected cells are strictly speaking not the appropriate control for the experiment transfecting TRAK1. Since a subtle redistribution of mitochondria is concluded from comparison of TRAK1 transfection to untransfected, the authors need to control for the effect of transfection alone vs the non-

transfected control. A simple comparison of the relevant metrics in non-transfected to control transfected cells should remove all concerns regarding the absence of this control, if indeed there is no effect of transfection alone.

We agree with the referee that because our analysis allows us to reveal small differences between conditions a reporter plasmid would better demonstrate that the transfection per se does not have non-specific effects on mitochondrial distribution. We have expressed GFP in WT and Miro^{DKO} cells and compared mitochondrial distribution between GFP expressing cells and untransfected cells from the same coverslips. We show that mitochondrial distribution is unaffected by a control (GFP) transfection. The Mito⁹⁵ value is very similar and the Mitochondrial Probability Maps (MPM) almost perfectly overlap between the untransfected and the transfected conditions. We comment on this finding prior to demonstrating the effect of overexpressed TRAK1/2 ± KIF5C on mitochondrial distribution and include this data in Appendix Fig S2A and B in the current version of the manuscript.

5) Fig 6A and C show an almost complete absence of myo19 by western in all cellular fractions analyzed by western and also by immunocytochemistry, respectively. This is very different from the interpretation initially presented in the text that in the absence of one or the other Miro there is less Myo19 at mitochondria, and almost none in the double KO. Rather, there seems to be a loss of myo19 in and throughout cells.

The authors arrive at this conclusion regarding the regulation of Myo19 levels in the next panels of Fig 6. However, as presented the reader gets the impression that the initial aspects of Fig 6 show specific effects on mitochondrial localization of myo19, which is not the case as the effect is on total myo19 levels and consequently resulting in decreased myo19 being present at mitochondria (and indeed elsewhere). Thus, it is suggested that the presentation of the data in Fig 6 be revised to first note that myo19 levels are impacted by Miro KD. Then, presentation of the experimental evidence for the regulation of myo19 levels, and finally that, as then predicted by the strong decrease in overall myo19 levels, there is correspondingly, if not expectedly, less myo19 at mitochondria. As presented, it seems a case of putting the cart before the horse.

This point has been similarly raised by all 3 reviewers. We apologize for not being clearer. Our hypothesis is that Miro proteins play a role in both Myo19 recruitment to mitochondria and Myo19 stabilization on the mitochondrial membrane. In the absence of Miro, Myo19 protein is less stable which explains the lower protein levels in Miro^{DKO} cells. In the revised version of the manuscript we include a significant amount of new data from experiments designed to more mechanistically address the relationship between Miro proteins and Myo19.

First, we perform a detailed quantification of the impact of Miro deletion on total Myo19 levels across all the genotypes. We show that total levels of Myo19 are decreased in Miro2^{KO} MEFs but also demonstrate a role for Miro1 in maintaining Myo19 levels because when both Miro proteins are deleted in Miro^{DKO} MEFs Myo19 levels are significantly further reduced.

Secondly, we present compelling evidence that Miro recruits Myo19 to the mitochondria. We show that exogenous overexpression of GFPMyo19 can localize in the mitochondria, which was already shown to occur through its MyMOMA domain (Hawthorne et al, 2016). We show that this happens even in the absence of Miro although the mitochondrial enrichment is substantially decreased in both Miro1^{KO} and Miro2^{KO} and even more so in Miro^{DKO} cells. Moreover, we show that mitochondrially localized Miro1 or Miro2 are able to rescue mitochondrial GFPMyo19 in Miro^{DKO} cells whereas in WT cells cytoplasmic Miro constructs lacking the transmembrane domain can increase the cytoplasmic signal of GFPMyo19 to levels similar to those found in Miro^{DKO} cells. This indicates that Miro has the ability to recruit and relocalize GFPMyo19 changing the balance between mitochondrial vs cytoplasmic pools.

Finally, we address whether the mitochondrial localization of Myo19 stabilizes the protein. We show that endogenous Myo19 presents two different pools. Myo19 is primarily mitochondrial in WT cells while there is a small pool of Myo19 localized in the cytoplasm. In Miro^{DKO} cells only the cytoplasmic pool is observed while the mitochondrial pool is almost completely lost. This observation suggests that the cytoplasmic pool is very unstable and that by recruiting Myo19 to the mitochondria Miro proteins stabilize Myo19 levels. In support of this hypothesis we now show that upon cycloheximide treatment (an inhibitor of protein synthesis) Myo19 levels in WT, Miro1^{KO} and Miro2^{KO} cells are stable over 12 hours treatment. In contrast, the remaining Myo19 levels (localizing only in the cytoplasm) are rapidly reduced in Miro^{DKO} cells treated with cycloheximide for the same time period. Furthermore, we show that Myo19 degradation is dependent on the proteasome but not on lysosomal degradation pathways.

We have rearranged the whole section and included this data in Fig 6 and in a new figure, Fig 7, in the revised form of the manuscript.

6) pg 20. The authors write that Miro plays a central role in coordinating "opposing" actions of microtubule and actin-dependent forces. It is not clear how the data show "opposing" forces/functions for microtubule vs actin based mechanisms. The second sentence in this paragraph seems a better description of the observations.

We have modified this section following the reviewer's recommendation to improve the clarity of the message we wanted to deliver.

Referee #2:

By generating Miro1/2 KO embryos and MEFs, López-Doménech et al (EMBOJ-2016-96380) report an unexpected role of mammalian Miro1/2 proteins in coordinating mitochondrial distribution through both microtubule- and actin-based motor mechanisms. This is a follow-up study from their recent work (López-Doménech et al., 2016) in cKO mouse models that show Miro1 as the primary regulator of mitochondrial transport in neuronal axons and dendrites. This study is also supported by an early mouse genetic study demonstrating that a neuron-specific loss of Miro1 causes the depletion of mitochondria from corticospinal tract axons and results in progressive neurological deficits (Nguyen et al., 2014). These studies call for an urgent investigation into whether mammalian Miro1 and Miro2 have overlapping or complementary functions and whether the total loss of Miro1/2 would still permit any mitochondrial transport in mammalian cells.

Using a substrate micro-patterning culture system, the authors generated a quantitative mitochondrial distribution map upon knockout of Miro1, Miro2 or both Miro1/2 in MEFs. By live imaging, they revealed that Miro1 appeared to be the primary mediator of mitochondrial trafficking along MTs. Surprisingly, they found that about 30% of MT-based mitochondrial movements were preserved in Miro1/2 DKO cells. The fractionation and imaging experiments further demonstrate that both kinesin and dynein motors, along with their motor adaptors TRAK1 and TRAK2, can still be associated with mitochondria despite a complete loss of Miro1/2. It is even more surprising that mitochondrial anterograde transport is enhanced by overexpression of TRAK and KIF5 in DKO MEFs. In addition, Miro proteins recruit Myo19 onto the mitochondrial outer membrane; these roles are critical for symmetric mitochondrial segregation during cell division. Based on these findings, the authors propose a new attractive model: Miro proteins play a central role in coordinating MT- and actin-dependent trafficking and positioning of mitochondria. These unexpected results challenge the current model that Miro proteins are the essential receptors for selectively recruiting kinesin-TRAK motor-adaptor complexes to mitochondria.

Overall, this is an interesting study, which could potentially make an important contribution to the mitochondrial transport field. The reported results are highly unexpected, thus, the study will fill a gap in our current knowledge and is suitable for the broad readership of the EMBO J. The manuscript is logically presented and most of the data are solid and in general of excellent quality. It represents a great deal of work, as evidenced by the comprehensive single and double KO analysis in mouse embryos and MEFs.

We thank the reviewer for their constructive comments which have significantly helped to improve our manuscript.

Main concerns:

1) The current study, however, is rather descriptive. It lacks cellular mechanistic insights into the observed phenotypes. For example: (1) How are TRAK1/2 recruited to mitochondria in the absence of Miro1/2? (2) How does Miro1 selectively facilitate TRAK2-dependent retrograde transport of mitochondria? (3) Is Miro-mediated recruitment / stability of Myo19 onto mitochondria sufficient to drive actin-based movement or actin-mediated anchoring of mitochondria? If this is the case, (4) how could these coupling defects impact mitochondrial segregation. Addressing one or two of these mechanistic questions would make this study more compelling not only to challenge the current models, but also to advance our knowledge of the Miro-TRAK-motor complexes for their coordination of mitochondrial trafficking and anchoring by engaging on MT and actin.

We agree that addressing one or two of the points raised would provide additional mechanistic insight that would further enhance the impact of the manuscript. Guided by the editor we have focused in particular on addressing how TRAK proteins may be stabilised on the mitochondrion in the absence of Miro and addressing in more depth the relationship between Miro and Myo19. We strongly feel this adds significant mechanistic insight to the manuscript. In addition we have also further investigated the impact of increasing Myo19 levels on the mitochondrion in the absence of Miro for mitochondrial segregation during cell division.

To address the mechanism by which TRAKs can still be recruited to mitochondria we have tested whether other mitochondrial candidates may be receptors of TRAK1/2 in the mitochondrial membrane. We have tested some candidates which have been already reported as regulators of mitochondrial transport. We have observed that syntaphilin or syntabulin are expressed at low levels in MEFs (which were undetectable in the mitochondrial fractions) suggesting they are unlikely to mediate TRAK binding in this system. Then we have focused on mitofusin1 (Mfn1) because it is known to interact with TRAK1 and TRAK2 (Lee et al, 2017; Misko et al, 2010). To test whether the Mfn1 / TRAK interaction is dependent on the presence of Miro proteins we have performed Proximity Ligation Assays (PLA), which detects *in situ*, protein interactions with high sensitivity and specificity. We show that TRAK1 can interact with Mfn1 even in the absence of Miro. Furthermore, we also describe an interaction between TRAK2 and Mfn1 which occurs more readily when Miro is absent. We have included these data in Fig EV4 and discussed it in the results and discussion sections.

We have also focused a significant part of our efforts in better understanding the mechanistic relationship between Miro and Myo19. We show that total levels of Myo19 are decreased in Miro2^{KO} MEFs but also demonstrate a role for Miro1 in maintaining Myo19 levels because when both Miro proteins are deleted in Miro^{DKO} MEFs Myo19 levels are significantly further reduced. We also show that exogenous

overexpression of GFPMyo19 can localize in the mitochondria, even in the absence of Miro, although the mitochondrial enrichment is substantially decreased in both Miro1^{KO} and Miro2^{KO} and even more so in Miro^{DKO} cells. Moreover, we show that mitochondrially localized Miro1 or Miro2 are able to rescue mitochondrial GFPMyo19 in Miro^{DKO} cells whereas in WT cells cytoplasmic Miro constructs lacking the transmembrane domain can increase the cytoplasmic signal of GFPMyo19 to levels similar to those found in Miro^{DKO} cells. This indicates that Miro has the ability to recruit and relocalize GFPMyo19 changing the balance between mitochondrial vs cytoplasmic pools. Finally, we show that endogenous Myo19 is primarily mitochondrial in WT cells while there is a small pool of Myo19 localized in the cytoplasm. In Miro^{DKO} cells only the cytoplasmic pool is observed while the mitochondrial pool is almost completely lost. This observation suggests that the cytoplasmic pool is very unstable and that by recruiting Myo19 to the mitochondria Miro proteins stabilize Myo19 levels. In support of this hypothesis we now show that upon cycloheximide treatment (an inhibitor of protein synthesis) Myo19 levels in WT, Miro1^{KO} and Miro2^{KO} cells are stable over 12 hours treatment. In contrast, the remaining Myo19 levels (localizing only in the cytoplasm) are rapidly reduced in Miro^{DKO} cells treated with cycloheximide for the same time period dependent on the proteasome but not on lysosomal degradation pathways.

Moreover by overexpressing GFPMyo19 we have been able to look at the consequences of restoring Myo19 without changing the levels of Miro in Miro^{DKO} cells. We report that overexpressed GFPMyo19 in Miro^{DKO} cells partially rescued the symmetric mitochondrial segregation of mitochondria. This demonstrates that Myo19 is in part responsible for the defects in mitochondrial segregation seen in Miro^{DKO} cells, but also indicating that Miro is still required to either regulate Myo19 activity or to regulate other process such as mitochondrial distribution along microtubules that are also required to ensure equal segregation of mitochondria during mitosis.

We hope that the reviewer will find these additions provide new mechanistic depth to the manuscript.

2) If the Miro proteins act as acceptors for Myo19, one should see the remaining Myo19 in the cytosol of DKO cells. It is difficult to interpret why almost all of Myo19 is lost in cell lysates and cytoplasmic fractions of DKO cells (Fig 6A and 6B).

This point is similar to point 5 raised by referee #1. We now demonstrate that the decreased levels of Myo19 in Miro^{DKO} MEFs are due to a reduced stability of cytoplasmic Myo19 in the absence of Miro. We show that Miro proteins are receptors of Myo19 in the mitochondria with the ability to recruit and stabilize Myo19 on the mitochondrion. The remaining levels seen in the cytosolic fraction of Miro^{DKO} cells (Fig 6C) are due to the equilibrium between protein degradation and protein synthesis. We demonstrate that, when not in mitochondria, Myo19 is very labile and is rapidly degraded by the proteasome which explains why the cytoplasmic levels are so low in Miro^{DKO} cells.

3) In addition, I am curious to know how the authors obtained the data showing the relative enrichment of Myo19 on mitochondria by calculating as mitochondrial signal / cytoplasmic signal. Again, the same concern applies to the imaging data (Fig 6C, 6D), where the Myo19 signal is lost globally in DKO cells, but not necessary to mitochondria-labeled Myo19. Based on these data, one would propose an alternative role of Miro in either regulating Myo19 expression or maintaining its stability.

As explained above (see point #5 of reviewer 1), in the revised version of the manuscript we have included a significant amount of new information supporting that Miro proteins recruit Myo19 to the mitochondria and by doing so stabilize Myo19 protein levels. We show that Myo19 levels are decreased in the absence of Miro but we want to point out that there is still a small quantity of Myo19 in Miro^{DKO} cells. Interestingly, this Myo19 happens to be cytoplasmic as can be seen in our fractionation experiments. Because the cytoplasmic pool of Myo19 is present in all the genotypes we conclude that there is a dynamic equilibrium between the cytoplasmic and the mitochondrial Myo19 which is affected by the ability of Miro to stabilize Myo19 in the mitochondria.

4) This new role of Miro is quite interesting considering its impact on actin-based movement versus anchoring, and thus presumably regulating mitochondrial segregation. However, the current study makes no effort to explore the possible mechanisms.

As explained above (point #1 of this reviewer) we have found that overexpressed GFPMyo19 in Miro^{DKO} cells partially rescued the symmetric mitochondrial segregation of mitochondria. This allow us to conclude that Miro proteins are important for regulating mitochondrial segregation through Myo19 stability. However, because mitochondrial segregation is not fully rescued even in the presence of overexpressed Myo19, Miro proteins are still necessary to ensure correct mitochondrial segregation during mitosis, either by further regulating Myo19 activity or by coordinating Myo19/actin related movement with microtubule dependent mitochondrial positioning/transport.

Other minor points:

5) Page 8: The statement: "The prevailing model of Miro function is that it acts as the essential adaptor for recruiting the motor complexes to the mitochondria to drive mitochondrial transport along the microtubule tracks". It would be more accurate to state: "...as the essential receptor for recruiting the motor-adaptor complexes.....".

We have modified this sentence as suggested.

6) Video 3: The majority of mitochondria in DKO are still in a tubular shape, which is not consistent with what is shown in Fig. 2A, 2B.

Fig 2A and B provide an analysis of mitochondrial morphology as part of our characterization of the novel Miro knockout cell lines. In mouse embryonic fibroblasts we observe a range (both within and across cells) of mitochondrial morphologies from long and tubular to shorter and more rounded. For the purposes of quantification we defined three morphological scoring groups (elongated, intermediate and short) and the three images provided are representative of these types of morphologies (not of the genotypes per se). Using these criteria we performed a blinded scoring analysis across many hundreds of cells for the different genotypes. In the analysis (panel 2B) about roughly 40% of Miro^{DKO} cells show a majority of mitochondria that are short while 35% of cells have mainly tubular mitochondria, whereas in wild type cells only a small number (less than 15%) score as having short mitochondria. We also defined an intermediate phenotype for cells where neither the tubular nor the short and rounded mitochondrial morphologies predominate. This group represented approx. 20-25% of cells in both WT and Miro^{DKO} genotypes. To address the reviewer's concern we have provided additional examples of the different mitochondrial morphologies observed in the Miro^{DKO} cells for the three scoring groups (as opposed to just showing the short and rounded Miro^{DKO} example). We have now provided an example video of

mitochondrial trafficking in a Miro^{DKO} cell with predominantly shorter mitochondria. Changes in mitochondrial morphology may be a consequence of defects in mitochondrial transport (which would likely reduce the number of collision events required for fusion) and we have mentioned this in the discussion section of our manuscript.

7) Fig 3 and Videos 1, 3: I am surprised by the quantitative data that shows such a small number of motile mitochondria associated with MT-based transport in each cell. In the video, I could easily see more mitochondria in WT cells undergoing such movement.

The relatively small number of mitochondria undergoing long range transport in our quantified data sets - compared to the apparent higher number of mitochondrial movements that can be observed in the videos – can be explained by the relatively restrictive parameters we used to define a motile mitochondrion. By taking into account only directional mitochondrial runs persistent over at least 5 microns we could ensure that we are primarily quantifying movements along the tubulin cytoskeleton and thus our values more accurately account for longer-range microtubule related transport events. This is indeed supported by the fact that using these parameters our analysis showed almost no microtubule related runs upon vinblastine treatment. However mitochondrial displacements where the actin- or microtubule-dependent contribution is harder to determine (such as lateral displacement, forward and backward changes in directionality or oscillatory movements), but which also contribute to the moving mitochondrial population observed in the movies will have been excluded from this particular analysis (although these mitochondrial displacements will have been represented in Fig 3A-B). We have made this clearer in the text and methods section.

8) Fig 4A: Fractions "I, C, M" should be noted in the fig legend. In Fig 4B, higher magnification images would be helpful to see the mitochondrial targeting of TRAKs.

We have included a definition for the labels in the legend for Fig 3 and 6 (which contains the mitochondrial fractionation data) and we have increased the magnification of the images showing TRAK recruitment to the mitochondria in Miro^{DKO} cells

Referee #3:

We thank the reviewer for their comments and helpful suggestions, which have significantly improved our manuscript.

The manuscript by Lopez-Domenech et al. investigates the role of Miro proteins in mitochondrial trafficking in MEFs generated from Miro1-, Miro2-, and double-knockout mice. The authors examine mitochondrial distribution in these MEFs, and the association of Trak1 and Trak2 as well as motors with mitochondria in these cells. Finally, the authors provide evidence that Miro1/2 double knockout interferes with the proper segregation of mitochondria to daughter cells during cytokinesis.

The generation of Miro knockout mice, first described in the paper from the same group last year (Lopez-Domenech et al., 2016) provides important models that are further investigated here. Previously, the authors studied the effects of Miro1 and Miro2 knockout on mitochondrial motility in neurons, here they focus on MEFs, and also on cells from double knockout animals.

1) The most interesting and novel observation from these follow-up studies is that both TRAK proteins

and microtubule motors are apparently still recruited to mitochondria in the absence of Miro proteins, suggesting that the canonical role for Miro in TRAK/motor recruitment is unlikely to be true.

We thank the reviewer for acknowledging that our findings that TRAKs and microtubule motors are still recruited to the mitochondria in the absence of Miro are interesting and novel and challenge the current model of mitochondrial transport. However, we strongly believe that the novel link between Miro proteins and the myosin motor Myo19 that we also describe in our manuscript are equally significant. To date, our knowledge on the mechanisms by which mitochondria are trafficked through the actin cytoskeleton remain less well understood. Our study provides new mechanistic insight on this matter and places Miro proteins center stage in the regulatory network that controls mitochondrial transport and distribution through the actin and the tubulin cytoskeleton with relevance to important cellular processes like cell division.

However, the first sentence of the abstract states that "Miro1 and Miro2 . . . regulate mitochondrial trafficking along microtubules by linking mitochondria to kinesin and dynein motors". This statement suggests that the authors do not yet believe their own data.

We do not feel that the semantics of how we nuanced the first sentence of our abstract justifies the above comment. In the very first sentence of the abstract we were simply aiming to summarize (as succinctly as possible due to the abstract word limit) the current knowledge in the field rather than the main findings of our new study. In the third sentence of our abstract we already state that TRAK1 and TRAK2 can not only still localize to mitochondria but can also be functional in the total absence of Miro. However to avoid ambiguity we have added the following to sentence 1 of the abstract: "In the current model of mitochondrial trafficking...".

2) Part of the problem may be that the authors examine a number of different aspects of mitochondrial trafficking, but do not go far enough in depth on any single point to make a clear and compelling case. Instead, the manuscript reads like two or three separate stories put together into a single manuscript. Overall the work feels fragmented with no single point explored in sufficient detail to make a cohesive story.

We respectfully disagree that our manuscript represents separate stories. From our experiments on mitochondrial displacement and trafficking in our MEF cells lines we arrived to two important conclusions that led the rest of our study. Our observation that tubulin-dependent trafficking of mitochondria in Miro^{DKO} cells was greatly reduced but not completely abolished led us to further study the impact of Miro loss on TRAK / kinesin-dependent trafficking. However our observation that an additional microtubule independent mitochondrial trafficking component was also disrupted lead us to focus on the actin cytoskeleton and more specifically Myo19, revealing the dual role of Miro in coordinating transport through both cytoskeletons. Moreover, it was known that Myo19 regulates segregation of mitochondria after cell division, a process known to also be dependent on microtubule regulation. Thus, we felt that this process was a good target of our investigation in demonstrating that the coordination of microtubule and actin-dependent mitochondrial positioning by Miro proteins is of physiological significance.

However, following suggestions raised by referee #1 (points 3 and 5) and referee #2 (points 1, 2, 3 and 4) in addition to this referee we have (as described below and in the preceding sections) further developed in more depth key aspects of the manuscript. We now show that at least mitofusin1 is able to interact and

recruit TRAK1 and TRAK2 to the mitochondrial membrane in the absence of Miro. Moreover we show that Myo19 is indeed depleted in Miro^{DKO} cells and that the reason for such depletion relies in the ability of Miro to recruit Myo19 to the mitochondrial membrane and stabilize it there. We also show that Myo19 can still localize in the mitochondria when overexpressed in the absence of Miro although to a lower extent than in the presence of Miro. Furthermore, we show that this partial enrichment of Myo19 in the mitochondrial membrane in the absence of Miro is not sufficient to completely rescue the equal segregation of mitochondria during mitosis indicating that additional functions of Miro proteins are still required to ensure an equal segregation of mitochondria after division. We strongly feel that the additional insight we provide in the revised version of the manuscript addresses the concerns of the reviewer regarding the perceived lack of depth in some aspects of our manuscript.

3) In addition, the manuscript is sloppy. The phrase "n=cells" is used, presumably as a placeholder for important statistical information that was never filled in.

We strongly disagree with the statement that our manuscript is 'sloppy'. We apologize for the confusion regarding the phrase "n= cells", which is not a placeholder that we forgot to fill in. As stated in the experimental procedures section (which provides an in depth description of analysis and statistics), "n= cells" was used to state that the 'n' number used to calculate error bars and perform statistics was the number of cells analysed. We have now included in the graphs or in figure legends the actual number of cells used in each condition for all the experiments where the number of transfected cells were used as the 'n' number.

4-1) Labels used in figures are not explained in the legends, such as I, C, and M in Figure 4.

We thank the reviewer for pointing this out (also raised by reviewer #2) and we have now included the I, C and M definitions in the figure legend. Please note, now the order of the fractions has changed (Input / Mitochondria / Cytoplasm) to make them consistent across all fractionation experiments.

4-2) Figure 1 is cut off at the top.

We have downloaded the file made for referees and in our hands this figure is not cut off on the top. Moreover this issue was not raised by the other two referees. We will consult with EMBO regarding any eventual formatting problem with the figure if this problem persists.

4-3) More significantly, the images shown in many panels do not appear to be representative of the overall quantitative conclusions - for example, compare panels C and G in Fig. 4. Additional specifics are given below.

We strongly disagree with the assertion that "images shown in many panels do not appear to be representative of the overall quantitative conclusions". When comparing the zoomed mitochondrial representative tip of each triangle image provided there is in fact a striking correspondence with the quantitative data provided from the Mito95. When for example comparing Fig 4 panel A to panel E (or 4B to H) or Fig 5 panel A to panel E (or 5B to H) in the current version of the manuscript, low mitochondrial signal in the zoom compares very closely to the lower values in the bar graph whereas high signal in the zoom corresponds to higher values in the bar graphs. We are therefore somewhat puzzled by the reviewers point which was not brought up by either of the other reviewers.

See also specific point F4-7 below.

In sum, this work is too preliminary for publication. Specific comments on Figures:

We respectfully disagree with the assertion that our original manuscript was too preliminary. Indeed neither of the other two reviewers suggested this and referee 2 pointed out that our original manuscript “represents a great deal of work, as evidenced by the comprehensive single and double KO analysis in mouse embryos and MEFs.”

Figure 1:

F1-1) • Panel A-C is cut off on the top.

We have downloaded the file made for referees and in our hands this figure is not cut off on the top. Moreover this issue was not raised by the other two referees. We will consult with EMBO regarding any eventual formatting problem with the figure and whether uploading a different figure format would resolve this if necessary.

F1-2) • It would be nice to have examples of M1het/M2KO, M2KO, and WT at all three developmental stages for comparison

Because the images were taken in a blinded fashion (i.e. prior to knowing the genotypes) it was not possible to preemptively display all the genotypes together. Double heterozygous pairs were set up to allow the generation of all possible genotypes and litters were obtained, examined, and imaged prior to genotyping (see Table 1). It is worth noting that the probability of an embryo to be WT or Miro2^{KO} is 6.25% for each of them (1/16). Due to space constraints we provided the most important comparative sets. At all developmental stages there is no difference in Miro2 embryo development compared to WT and therefore Miro2^{KO} serves as an appropriate control. Indeed, the comparisons of Miro1^{het}/Miro2^{KO} embryos with Miro2^{KO} embryos highlights the importance of having 2 copies of Miro1 when there is no Miro2 in the embryo. We have included the following sentence in the results section to make the point clearer. “In contrast, embryos with only one allele of Miro1 (Miro1^{het}/Miro2^{KO}) were only found to be viable until E12.5 indicating that having only one copy of Miro1 is not enough to compensate the lack of Miro2 beyond E12.5 (Table 1 and Fig 1A - C)”. We have also clarified this in the figure legend. However we also provide, in Fig EV1A, an example of a WT and a Miro2^{KO} embryo at E16.5

F1-3) • Panel E the scale bar is mislabeled (presumably that is not 500mm) and the arrows are not described in the figure legend

We thank the reviewer for noticing this error. We have re-labelled this scale bar and described arrows in the figure legend.

F1-4) • In Panel F the order of mutants is confusing and is not consistent between blots F and G. This certainly does not invalidate the findings, but it makes it very difficult to follow.

We understand the point the reviewer is making and apologise that the gel is a little hard to follow but as already pointed out by the reviewer the order that the samples were run in does not invalidate the findings. Although the order may be a little harder to follow we would like to keep these gels as they are in the figure because they are part of the gels that were actually quantified for Fig EV1.

Figure 2:

F2-1) • Throughout the manuscript the authors describe using two different cell lines per genotype. While three independent experiments are carried out on these MEFs, this is nonetheless only two biological replicates.

~~We are really unclear as to the point the referee is trying to make. We have performed the experiments with two independently generated MEF cell lines per genotype which were found to show the same behavior. These experiments were performed at least 3 times per genotype using 10 to 20 different cells per condition and per experiment. Our use of replicates, experiments and “n” numbers for statistics are clear and, we believe, well justified. To our knowledge most papers use only one MEF cell line per genotype. However, we decided to independently generate 2 different MEF cell lines for every single genotype used throughout the manuscript to allow us to confirm that our findings are not due to MEF cell line variability. We honestly believe that this should be considered a strength in our experimental design instead of being a target of criticism.~~

F2-2) • The authors should consider using a color scheme more friendly to colorblind readers.

We thank the reviewer for this suggestion. We have changed the green/red pairs of images for a more appropriate (magenta/green) color scheme for colorblind readers.

F2-3) • Figure C does not have a scale bar in the detail column

We have added this scale bar. We have also added a similar scale bar in Fig 4A and B and Fig 5A and B in the detail images.

Figure 3:

F3-1) • The technique used to describe mitochondrial displacement is not clearly described in the legend

We have made a clearer description of mitochondrial displacement analysis in the figure legend. A full description is provided in the methods section.

F3-2) • In panel A, the double knockout cell does not appear to have the perinuclear clustering phenotype described in figure 2

We respectfully disagree with the reviewer on this point. What we show is that the Miro^{DKO} cells exhibit a significantly more accumulated mitochondrial distribution towards the cell nucleus. Importantly, it is not possible to make conclusions (as the reviewer has done) of the extent of mitochondrial perinuclear distribution in the cells presented in Fig 3 (either WT or DKO) as the cells were not co-expressing any cell fill reporter that would allow a delineation of the cell periphery. Given the heterogeneous nature of MEF cell morphology one cannot easily conclude on the extent of peri-nuclear mitochondrial accumulation in those cells. It is specifically for this reason that our conclusions on the impact of Miro knockout on mitochondrial distribution are derived from a mitochondrial distribution assay performed on cells grown on micro-patterned substrates which was designed to accurately analyze mitochondrial distribution in a way that is not affected by cell size or morphology (two parameters that are variable in

fibroblasts freely growing in a dish) and that also allows the cell periphery to be accurately delineated by expression of a cell fill reporter or by immunostaining.

F3-3) • In panel C the mitochondria in the double knockout cell do not appear to display the fragmentation described in figure 2

We have not described mitochondrial morphology as being fragmented in Miro^{DKO} cells. The phenotype we report is that of shorter mitochondria (possibly as a consequence of reduced trafficking). However Referee#3 does also raise here an issue raised by referee #2. As also pointed out in our response to referee #2, Fig 2A and B provide an analysis of mitochondrial morphology as part of our characterization of the novel Miro knockout cell lines. In mouse embryonic fibroblasts we observe a range (both within and across cells) of mitochondrial morphologies from long and tubular to shorter and more rounded. For the purposes of quantification we defined three morphological scoring groups (elongated, intermediate and short) and the three images provided are representative of these types of morphologies. Using these criteria we performed a blinded scoring analysis across many hundreds of cells for the different genotypes. In the analysis (panel 2B) roughly 40% of Miro^{DKO} cells show a majority of mitochondria that are short and rounded while 35% of cells have mainly tubular mitochondria, whereas only a small number of wild type cells (less than 15%) score as having short mitochondria. We also defined an intermediate phenotype for cells where neither the tubular nor the short and rounded mitochondrial morphologies predominate. This group represented approx. 20-25% of cells in both WT and DKO genotypes. Thus about 55% of Miro^{DKO} cells do not show a short mitochondrial phenotype, which explains why some Miro^{DKO} cells (such as the one in Fig 3C) do not have short mitochondria. To address the reviewer's concern we have provided additional examples of the different mitochondrial morphologies observed in the Miro^{DKO} cells representative of the three scoring groups (as opposed to just showing the short and rounded Miro^{DKO} example).

F3-4) • In panel D Vinblastine is incorrectly spelled as "vinblastin" and the concentration reads "µm" instead of µM.

We thank the reviewer for pointing out these mistakes which we have now corrected.

F3-5) • Again, in panel F the mitochondria in the DKO column appear more elongated than the mitochondria in the WT column (in contrast to the claims from fig 2.).

As explained above (see point F3-3 from this referee and point 6 raised by referee #2), about 55% of Miro^{DKO} cells do not show a short mitochondrial phenotype. To better represent all the range of mitochondrial morphologies in Miro^{DKO} cells we have provided additional examples of cells with different mitochondrial morphologies observed in the Miro^{DKO} cells representative of the three scoring groups (as opposed to just showing the short and rounded Miro^{DKO} example).

Figure 4:

F4-1) • This figure demonstrates a new and potential exciting observation. Namely, Trak1, Trak2, and KHC are still recruited to mitochondria in the absence of Miro1 and 2. However, no quantitative analysis is provided.

F4-2) • Importantly, these proteins not only seem to be recruited to the mitochondria, but their localization to mitochondria appears to be unregulated in the mutants.

F4-3) • However, the authors fail to address the significance of the apparent upregulation of Trak1 and KHC in the mitochondrial fraction of the M1KO M2KO and DKO cells in panel A.

F4-4) • This is particularly important, as the subsequent experiments in figure 4 and 5 rely on overexpression of exogenous forms of these proteins on top of already altered protein levels in the mitochondria of the various mutants.

F4-5) • Quantitation of the western blot from at least three independent biological replicates would be informative, in order to clarify TRAK1/2 and KIF5c levels on mitochondria.

This point was also raised by reviewer #1 (See point 3). We have quantified the mitochondrial enrichment of all adaptor and motor proteins shown in the figure (now Fig 3E and F in the current version) from 4 independent fractionations. Our analysis shows that none of the motor/adaptor proteins tested shows any significant difference in their mitochondrial levels.

F4-6) • The authors should consider depleting endogenous TRAK1/2 in the mutant cell lines

We thank the reviewer for this suggestion. However we already present an in depth analysis of multiple parameters of mitochondrial distribution and trafficking in a series of Miro deleted cell lines. In addition we also look in detail at the relationship between Miro proteins and Myo19. We therefore believe that additionally depleting TRAK proteins across our Miro cell lines lies beyond the scope of the current manuscript.

F4-7) • In panels C-G, it appears as though the representative images in C do not convincingly reflect the quantitative data in G. For example, the TRAK2 column in panel C is almost completely absent of mitochondria, while the TRAK2 + KIF5C representative image shows clear mitochondria in the enlarged inset. This appears to be at odds with panel G, which indicates no significant difference between these two conditions. This casts doubt on several of the effects reported.

We strongly disagree with the assertion that images shown do not reflect the quantitative data. When comparing the zoomed mitochondrial image representative tip of each triangle provided there is in fact a striking correspondence with the quantitative data provided from the Mito95 in the vast majority of panels shown. When for example comparing Fig 4 panel A to panel E and B to H and Fig 5 panel A to panel E and B to H in the current version of the manuscript, low mitochondrial signal in the zoom compares very closely to the lower values in the bar graph whereas high signal in the zoom corresponds to higher values in the bar graphs.

We do however, agree with the reviewer that for the one specific comparison that they mention (TRAK2 vs TRAK2 + KIF5C) the correspondence is less obvious and thank them for bringing this to our attention. However we would like to point out that while only one zoomed tip of the reference cell triangle is selected for presentational purposes the quantitative analysis reflects the average of the 3 tips of the reference cell triangle. We have now provided another tip from the same reference cell (by turning the cell anticlockwise) that is more representative of the quantitative data.

Figure 6:

F6-1) • Again, this is interesting that myosin 19 fails to bind to mitochondria in the absence of Miro1/2. Further, panel E nicely demonstrates that rescue with either Miro can recover Myo19 localization to mitochondria

F6-2) • It would be nice if the authors discussed the Miro dependent mitochondrial localization of Myo19 in light of the findings from Hawthorne et al. (2016, Cytoskeleton) indicating that Myo19 is targeted to the mitochondrial outer membrane by positively charged residues within its MyMOMA domain.

As described above we have further investigated the mechanisms leading to altered localization/stabilization of Myo19 in mitochondria by Miro1 and 2 and consequences of this (see also referee#1 point 3 and referee#2 points 1 to 4). We have shown that Myo19 can still be targeted to the mitochondria in the absence of Miro indicating that the Myo19 MyMOMA domain is sufficient for the mitochondrial targeting but not enough to stabilize Myo19 in the mitochondria as demonstrated by the relatively low mitochondrial enrichment and the high levels of Myo19 in the cytoplasm of Miro^{DKO} cells overexpressing GFPMyo19. We have discussed these findings in terms of the MyMOMA domain in the discussion of the revised manuscript.

F6-3) • Mito tracker is spelled wrong in panel E

We have corrected this.

F6-4) • Panels H-I appears to be the only evidence substantiating the claim that "Moreover, Miro depletion during PINK1/Parkin-dependent mitophagy can also drive a loss of mitochondrial Myo19 upon mitochondrial damage."

F6-5) • While the authors show that Myo19 and Miro1 have different degradation kinetics upon FCCP treatment, this does not indicate that Miro depletion is responsible for the loss of Myo19.

While we agree that the panel doesn't prove that Miro depletion is responsible for the loss of Myo19 our aim was to use a paradigm where we could look at the consequences of rapid depletion of Miro1 (which has been established by several groups to occur rapidly upon mitochondrial damage) for the stability of Myo19. In the revised version of the manuscript we clearly show that Myo19 levels are dependent on Miro protein content. Moreover, we show that Miro proteins are able to recruit and stabilize Myo19 to the mitochondria. When there is no Miro, Myo19 levels are primarily cytoplasmic and very unstable as shown by the fast rate of Myo19 degradation in conditions where synthesis of new protein is blocked. We agree that we do not provide direct evidence that Miro depletion is the reason why Myo19 is degraded (for this we would need to be able to selectively block Miro degradation by Parkin which we currently cannot do) but instead points to a direct correlation supported by our analysis of recruitment/stability of Myo19 to the mitochondria by Miro proteins. Therefore we propose to change the wording of the sentence to make clear that the observation points to such a correlation suggesting that Miro degradation might be necessary to allow for Myo19 degradation during a mitophagic process.

To show this more clearly, the authors should examine the rate of FCCP driven myo19 degradation in Miro1 KO, and Miro 2 KO cells.

F6-6) • It would also be nice to see the experiments in H and I performed in the MEFs described throughout the paper

We would like to point out that our FCCP experiment was carried out in an SH-SY5Y cell line stably overexpressing Parkin (Birsa et al, 2014) where we previously performed a detailed characterisation of Miro degradation kinetics. We chose this cell line due to the rapid parkin-dependent degradation of Miro we previously observed in these cells. While we are grateful for the suggestion we do not feel that performing the FCCP experiments in the various Miro MEF lines would allow an easy comparison with the

SHSY5Y experiments and therefore we feel these experiments fall outside of the scope of the current work.

F6-7) • Additionally, the authors should examine the degradation kinetics of other outer membrane markers such as TOMM20 or Mitofusin 1.

We have performed the mitofusin1 and Tom20 blots as requested. Mfn1, as Miro1, undergoes rapid degradation under FCCP treatment as has been previously shown. In addition, Tom20 is also rapidly degraded upon FCCP treatment but with much slower kinetics as previously described for mitochondrial proteins located in the OMM (Birsa et al, 2014; Cohen et al, 2008; Yoshii et al, 2011). We include a representative blot below together with the western blots that originally appeared in the figure. However, we feel that these additional panels may detract from the main point we want to deliver here, this being, Myo19 levels undergo rapid decrease after mitochondrial damage with kinetics that closely follow degradation of Miro1.

F6-8) • Finally, the experiment should be repeated in the presence of a proteasome inhibitor such as MG132 to determine whether Myo19 and Miro degradation is driven by the proteasome or autophagic degradation.

We have investigated whether Myo19 destabilization and decreased levels are due to the targeting of Myo19 to the proteosomal or lysosomal/autophagic pathway. We include a detailed characterization of the degradation rate of Myo19 in our MEF cell lines with different content of Miro proteins. Furthermore we also show that Myo19 degradation in the absence of Miro proteins is dependent on the activity of the proteasome and not by a lysosomal/autophagic process. The results are now compiled in Fig 7 in the revised version of the manuscript.

Figure 7:

F7-1) • The authors show that mitochondria are unequally segregated between daughter cells in the Miro1/2 DKO. This observation suggests that at least one Miro is required for appropriate mitochondrial network distribution during cytokinesis.

F7-2) • What are the functional consequences of this asymmetrical distribution of mitochondria?

We believe that a consequence of an asymmetric segregation of mitochondria is reflected in the lower mitosis rate of Miro^{DKO} cells. We have also observed an increased death rate in Miro^{DKO} cells although the number of cells that died during our long term imaging experiments is very low, and even lower is the number of cells dying that previously went through a mitotic process during the movies. This prevented us from investigating whether there was any correlation between the unequal segregation of mitochondria and a later cell death event. However, we discuss in the revised version of the manuscript that the unequal segregation of mitochondria could increase the probability of cell cycle arrest and cellular death *in vivo*.

F7-3) • Do you observe recovery of mitochondrial mass in the cells that receive a smaller proportion of mitochondria from the mother cell?

We believe that this is an interesting suggestion. Our data suggests that this doesn't happen as we already described a lot of variability in mitochondrial content in Miro^{DKO} cells (Fig 8A and B).

F7-4) • Do you observe an increase in multinucleate cells in the DKO cells similar to what was described by Rohn et al (2014 Current biology) in Myo19 KD cells.

We have pursued this question by analyzing the quantity of bi-nucleated cells in the different MEF lines. We have observed that there is not any difference in the proportion of binucleated cells between WT and Miro^{DKO} cells (% of binucleated cells, WT: 3.72 ± 1.22 ; Miro^{DKO}: 5.50 ± 2.69 ; $p=0.590$, t-test. Data collected from 3 different experiments). We note that the % of binucleated cells is significantly lower than that reported when Myo19 is knocked down (Rohn et al, 2014). This can be due to the nature of the cell lines used, transformed MEFs in our work or neuronal tumor cell line (CAD) in Rohn et al.

F7-5) • Throughout the paper sample sizes are not properly indicated (often n values just say n=cells)

We have included all n values in figures and/or figure legends

F7-6) • For the SHOLL analysis, the authors do not report the radii of each concentric circle, nor do they describe the average radii of the nuclei of each cell.

We provide the radii increment for each concentric circle in page 30 in experimental procedures of the manuscript. We have also applied the Sholl analysis to provide an average distance where 95% of the nuclei is found in our experiments and we have included this distance in Fig 2D to give an idea on where the nuclei is found. We have also compared the nuclei size between WT and Miro^{DKO} cells. This data is in Fig EV3G

References:

Birsa N, Norkett R, Wauer T, Mevissen TE, Wu HC, Foltynie T, Bhatia K, Hirst WD, Komander D, Plun-Favreau H, Kittler JT (2014) Lysine 27 ubiquitination of the mitochondrial transport protein miro is dependent on serine 65 of the parkin ubiquitin ligase. *The Journal of biological chemistry* **289**: 14569-14582

Cohen MM, Leboucher GP, Livnat-Levanon N, Glickman MH, Weissman AM (2008) Ubiquitin-proteasome-dependent degradation of a mitofusin, a critical regulator of mitochondrial fusion. *Molecular biology of the cell* **19**: 2457-2464

Hawthorne JL, Mehta PR, Singh PP, Wong NQ, Quintero OA (2016) Positively charged residues within the MYO19 MyMOMA domain are essential for proper localization of MYO19 to the mitochondrial outer membrane. *Cytoskeleton* **73**: 286-299

Lee CA, Chin LS, Li L (2017) Hypertonia-linked protein Trak1 functions with mitofusins to promote mitochondrial tethering and fusion. *Protein & cell*

Misko A, Jiang S, Wegorzewska I, Milbrandt J, Baloh RH (2010) Mitofusin 2 is necessary for transport of axonal mitochondria and interacts with the Miro/Milton complex. *The Journal of neuroscience : the official journal of the Society for Neuroscience* **30**: 4232-4240

Rohn JL, Patel JV, Neumann B, Bulkescher J, McHedlishvili N, McMullan RC, Quintero OA, Ellenberg J, Baum B (2014) Myo19 ensures symmetric partitioning of mitochondria and coupling of mitochondrial segregation to cell division. *Current biology : CB* **24**: 2598-2605

Yoshii SR, Kishi C, Ishihara N, Mizushima N (2011) Parkin mediates proteasome-dependent protein degradation and rupture of the outer mitochondrial membrane. *The Journal of biological chemistry* **286**: 19630-19640

Thank you for submitting a revised version of your manuscript. The manuscript has now been seen by the three original referees, who find that all their main concerns have now been addressed. There are just a few minor editorial issues to be dealt with before formal acceptance here. Congratulations on a nice study!

REFeree COMMENTS

Referee #1:

The manuscript provides novel insights into the role of Miro1/2 in the regulation of mitochondrial biology. The authors were very responsive to the initial review and have revised the manuscript accordingly and provided new lines of evidence. This reviewer has no additional concerns.

Referee #2:

Revision by López-Doménech et al (EMBOJ-2016-96380R) has added an analysis to address my main concerns, thus building a strong case for their conclusions, which could make an important contribution to the mitochondrial transport field. I recommend that the current revision is suitable for publication in EMBO J.

Referee #3:

The authors have addressed the points raised in the initial review sufficiently so that I now recommend that this work be published in EMBO.

Corresponding Author Name: Josef T. Kittler

Manuscript Number: EMBOJ-2016-96380